# Bio-inspired asymmetric Zn-N₂O₂ single-atom catalysts via natural skeleton for efficient N-alkylation of nitroarenes with alcohols

Yu Huang [1], Yan Li[2], Xiaogang Yin [1] ✉, Qiudi Zhu[1], Mei He[1], Xueqin Chang[1], Xuefei Liu[1], Wu Li [2], Aiwen Lei [2] ✉ & Xianglin Pei [1] ✉

Although significant developments are made in non-noble metal catalysts for N-alkylation of nitroarenes with alcohols via borrowing hydrogen strategy, obtaining catalysts with superior activity, reusability and broad substrate scope under mild reaction conditions remains challenging. Single-atom catalysts (SACs) hold unique coordination/electron structures, to be the potential candidates for this reaction. In this study, we firstly and creatively fabricate bio-inspired Zn SACs with asymmetric Zn-N₂O₂ sites by utilizing the natural skeleton of biomass chitosan (denoted as Zn/CS), and achieve the first instance of heterogeneous Zn SACs in borrowing hydrogen reaction between nitroarenes and alcohols. The results reveal that the asymmetric Zn-N₂O₂ sites induced by natural skeleton (like ligands) and nanoporous structure of Zn/CS significantly promote the N-alkylation efficiency of nitroarenes with alcohols. Notably, the Zn/CS exhibits the highest turnover frequency (TOF) among the reported heterogeneous catalysts, as well as wide substrate scope (56 examples) and excellent reusability. Furthermore, the catalytic pathway/mechanism is investigated by combing theoretical calculations, which reveals that the asymmetric Zn-N₂O₂ sites with electron-deficient character can facilitate the formation of Zn-H and Zn-O bonds between Zn/CS and Ph-CH₂O⁻, thus easily generating the transition state Ph-CH₂O* and driving the whole reaction.

In recent years, N-alkylation compounds have become ubiquitous in pharmaceuticals[1], insecticides[2], additives[3], organic synthesis intermediates[4], functional materials[5], etc. Consequently, the development of green and efficient synthetic methods for N-alkylation compounds have emerged as a crucial goal in modern catalytic research. Traditional methods for synthesizing N-alkylation compounds, included Buchwald–Hartwig coupling, Hofmann degradation, Ullmann or Goldberg coupling reactions, etc[6–9]. However, these methods suffered from the several drawbacks: (1) the use of toxic alkylating reagents of

organohalides, and the wide range of by-products; (2) the excessive utilization of hazardous additives, such as silanes, boranes, or CaH₂; (3) the harsh reaction conditions, such as high temperature and pressure; (4) lower reactivity and applicability, which significantly restricted their applicability. Recently, borrowing hydrogen reactions catalyzed by transition metals employing alcohols and amines have demonstrated substantial advantages in green and efficient synthesis of N-alkylation compounds. These advantages include: (1) alcohol substrates, which are inexpensive, readily available in nature, and can

[1]School of Chemistry and Materials Science, Guizhou Normal University, Guiyang, China. [2]College of Chemistry and Molecular Sciences, Wuhan University, Wuhan, China. ✉e-mail: m13885115516@163.com; aiwenlei@whu.edu.cn; xianglinpei@163.com

serve as alkylating reagents, replacing toxic organohalides or halogenated heavy metal salts; (2) the reaction does not require harmful reducing additives, as the hydrogen comes from the alcohol itself; (3) the reaction occurs via a one-pot tandem process, and the by-product is only $H_2O$, offering high atom economy and environmental friendliness. Furthermore, researchers have explored using more cost-effective nitroarenes instead of amines as starting materials, attempting to improve the economic and environmental friendliness of N-alkylation compound synthesis, and have made a series of research progress[10,11].

In synthesizing N-alkylation compounds via the borrowing hydrogen strategy between nitroarenes and alcohols, the alcohols need to be oxidized to aldehydes, and act as hydrogen donors. Then it not only effectively reduces the nitroarenes to amines, but also the imine intermediates formed by aldehydes and amines to the desired N-alkylation compounds. Thus, this reaction has stringent requirements for the catalysts. Although numerous studies on homogeneous catalysis have reported the use of transition metal catalysts for N-alkylation reactions, which are constrained by issues, such as product contamination, difficulty in separation, and the high cost of ligand synthesis. Therefore, it is critical to develop green, efficient, and recyclable catalysts for the synthesis of N-alkylation compounds[12-14]. In recent years, a series of supported catalysts based on metals, such as Pd, Pt, Au, Ru, and Ir were successfully employed in borrowing hydrogen reactions between nitroarenes and alcohols[15]. Nevertheless, most of these reports focused on the noble metal catalysts, and the high cost and rarity limited their widespread application. In some reported non-noble metal catalysts, such as some reported carbon nanotube/Co, $Cu/Al_2O_3$, $Fe_3O_4@N-C$, etc., the conversion of the target product could be achieved[16-18]. However, due to the intrinsic activity of non-noble metals, they tended to exhibit poor catalytic performance or harsh reaction conditions, such as higher reaction temperatures, larger catalyst amounts, narrower substrate scopes, etc., which was contrary to green chemistry and limited their catalytic potential. It can be seen that achieving an efficient and green borrowing hydrogen reaction between nitroarenes and alcohols under mild conditions by using supported non-noble metal catalysts remains a significant challenge.

Chitosan, as the deacetylation product of the second largest natural polymer chitin in nature, is a green, low-cost, and renewable material. As a catalyst support, chitosan offers the several advantages: (1) its molecular chain contains high-density functional groups(-OH, $-NH_2$), allowing it to regulate the coordination environment and electronic structure of metal sites like ligands in homogeneous catalysts[19,20]; (2) due to its inherently rigid molecular structure, chitosan possesses a natural hierarchical architecture and can be further modulated through a sol-gel strategy, as well as excellent reusability, making it potential candidates for heterogeneous support; (3) the hierarchical architecture and N or O elements/groups in chitosan can promote the attachment and dispersion of metal particles, as well as the diffusion and exchange of reaction substrates. These factors indicate that using chitosan as a support for developing high-performance heterogeneous catalysts for N-alkylation of nitroarenes with alcohols holds considerable promise.

In this work, we dissolved chitosan in a low-temperature alkali/urea aqueous system and prepared porous three-dimensional (3D) chitosan microspheres via a sol-gel method. These microspheres exhibited larger surface area, which further facilitated the anchoring of metal particles and the rapid exchange of reactants. Furthermore, using these chitosan microspheres as support, we firstly successfully synthesized the bio-inspired Zn SACs and achieved the first application of Zn SACs in the borrowing hydrogen reaction between nitroarenes and alcohols. The study revealed that the Zn SACs supported by chitosan skeleton exhibited an asymmetric $Zn-N_2O_2$ coordination environment and electronic structure, which significantly enhanced

the catalytic performance of Zn/CS. Notably, the Zn/CS achieved the highest TOF with $51.85 h^{-1}$ compared to various reported heterogeneous catalysts, as well as excellent substrate compatibility (56 examples) and recycling stability. Combined with density functional theory (DFT) calculations, the corresponding catalytic mechanism was investigated, which indicated that the bio-inspired $Zn-N_2O_2$ sites with electron-deficient character significantly promoted the rate-determining step (RDS) of dehydrogenation conversion of alcohols, unveiling the fundamental reasons for the high catalytic activity of Zn/CS. This research utilizes the natural skeleton of biomass chitosan to synthesize bio-inspired Zn SACs for the first time, and achieves the first instance of heterogeneous Zn SACs for efficient catalysis of N-alkylation, which provides a new approach for efficient synthesis of N-alkylation compounds and contributes to the development and utilization of biomass resources.

## Results
### Structure characterization of Zn/CS
The chitosan powder was dissolved in an aqueous mixture of LiOH/KOH/urea using low-temperature freeze-thaw technology to obtain a transparent chitosan solution. Then, a sol-gel strategy was employed to recombine the hydrogen bonds within the chitosan molecules to form nanoporous chitosan microspheres via the emulsion method. Subsequently, the above chitosan nanoporous microspheres were impregnated with $Zn(NO_3)_2 \cdot 6H_2O$ to produce the $Zn^{2+}/CS$ precursor catalyst, which was then activated at 250 °C in an Ar atmosphere to obtain the final Zn/CS catalyst. The illustration for the preparation of Zn/CS catalyst was displayed in Fig. 1a.

To clarify the structure of the Zn/CS composite, various characterizations were performed. Scanning electron microscopy (SEM) images in Figs. 1b, c, and S1 demonstrated that both pure chitosan and Zn/CS exhibited uniformly distributed microspheres with 3D nanoporous structures, which facilitated the attachment and dispersion of Zn particles. SEM-EDX images in Fig. S2 confirmed the presence of C, N, O, and Zn elements in the Zn/CS catalyst, with Zn element uniformly distributed on the chitosan microspheres, demonstrating the successful loading of Zn particles. Nitrogen adsorption/desorption isotherms in Fig. 1d and Table S1 revealed that both the pure chitosan microspheres and Zn/CS all exhibited the type IV with H3 hysteresis loops, further indicating the mesoporous structure of materials, with pore sizes ranging from 1 to 100 nm, and a primary pore size of approximately 3.68 nm (inset in Fig. 1d)[21,22]. The pure chitosan microspheres and Zn/CS had large specific surface areas of 186.36 and 194.07 $m^2/g$, respectively. The large surface area could provide abundant attachment sites for Zn particles, as well as the diffusion of reactants.

Fourier transform infrared spectroscopy (FT-IR) results in Fig. S3 showed that compared to the pure chitosan, the synthesized Zn/CS catalyst retained the characteristic structural features of chitosan, such as: $-C-O-C$ (1081 $cm^{-1}$), $-NH_2$ (1620 $cm^{-1}$), $-NH-C=O$ (1697 $cm^{-1}$), $-OH$ (3446 $cm^{-1}$)[23]. X-ray diffraction patterns (XRD) depicted in Fig. S4 revealed that the Zn/CS catalyst maintained the characteristic diffraction peak of chitosan at 20.34°, further indicating the good stability of the chitosan support[24]. Notably, the Zn/CS catalyst did not exhibit the characteristic peaks of Zn, which may be attributed to the high dispersion of Zn particles on chitosan[25]. X-ray photoelectron spectroscopy (XPS) in Fig. S5 further verified the successful loading of Zn[26]. Specifically, the peaks at 1021.87 and 1045.06 eV appeared in Fig. 1e corresponded to the Zn $2P_{2/3}$ and Zn $2P_{1/2}$ peaks[27], respectively, which suggested that Zn existed in a $Zn^{+\delta}$ state and confirmed the absence of Zn nanoclusters[28-30]. As depicted in Fig. S6, with the C1$s$ peak calibrated at 284.80 eV, the C1$s$ spectra of the pure calcined chitosan and Zn/CS all displayed peaks at 284.80, 286.22, and 287.87 eV, corresponding to the C-C/C-H, C-O/C-N, and C=O/C=N bonds, respectively, suggesting no interaction between Zn and C

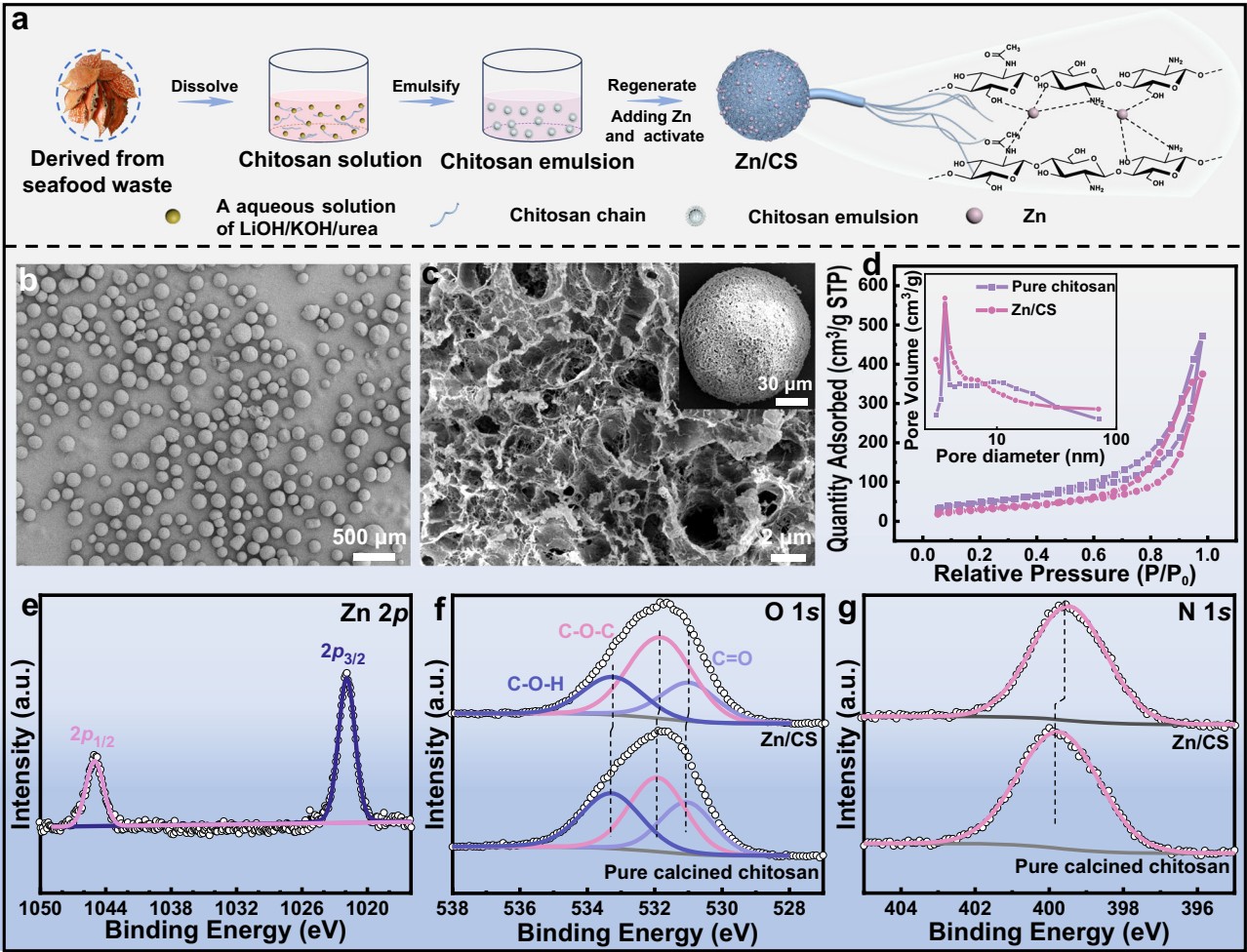

**Fig. 1 | Preparation and structure of the Zn/CS catalyst. a** Schematic diagram of the preparation process of Zn/CS catalyst. **b** SEM image of the Zn/CS catalyst. **c** Partial enlarged view of individual chitosan microspheres, inset it with a single chitosan microsphere. **d** Nitrogen adsorption/desorption isotherms of the pure chitosan and Zn/CS catalyst. **e** XPS spectra of Zn 2*p* for the Zn/CS catalyst, **f** O1*s* and **g** N1*s* spectra for the pure chitosan and Zn/CS catalyst.

element[31]. After the introduction of Zn, the binding energies of O1*s* (C=O, C–O–C, and C–O–H) in Zn/CS catalyst shifted slightly from 531.05, 531.92, and 533.29 eV to 530.38, 531.83, and 533.24 eV, respectively (Fig. 1f)[32]. Additionally, the N1*s* binding energy in Zn/CS shifted from 399.75 to 399.69 eV compared to the pure calcined chitosan (Fig. 1g)[31]. These data confirmed the interaction between N/O elements and Zn atoms, and the electrons transfer from Zn to the N/O atoms in chitosan[33]. Inductively coupled plasma optical emission spectrometry (ICP-OES) data indicated that the Zn content in Zn/CS catalyst was 1.11 wt%.

However, transmission electron microscopy (TEM) images in Figs. 2a and S7 showed no observable Zn particles[34,35]. Further analysis was conducted using high-angle annular dark field scanning transmission electron microscopy (HAADF-STEM), which revealed that numerous single bright spots were uniformly distributed on the chitosan support without aggregation, indicating the existing of Zn single atoms (Fig. 2b). Additionally, elemental mappings of HAADF-STEM further confirmed that Zn atoms were uniformly dispersed throughout the chitosan support (Fig. 2c)[36].

The coordination environment and electronic structure of the Zn/CS catalyst were further investigated by using X-ray absorption near-edge structure (XANES) and extended X-ray absorption fine structure (EXAFS) analyses. As shown in Fig. 2d, the XANES spectra revealed that the Zn K-edge position of the Zn/CS catalyst was close to that of standard references ZnO and ZnPc, but significantly different from

that of Zn foil[37]. It indicated that the valence state of Zn was between 0 and +δ, which was consistent with the XPS of Zn 2*p* analysis[38]. The EXAFS results in Figs. 2e, f and S8–11 revealed a prominent peak at 1.99 Å in the Zn/CS catalyst, similar to those observed in ZnPc and ZnO, which was corresponded to the Zn–N or Zn–O bonds with coordination number of approximately 4[39]. No characteristic peaks for Zn–Zn bonds were observed, confirming the absence of Zn nanoparticles, which was consistent with the HAADF-STEM and TEM observations. The detailed fitting data of the EXAFS spectra were summarized in Table S2. Wavelet transform (WT) analysis was also employed to further investigate the coordination structure of Zn/CS catalyst. As depicted in Fig. 2g–j, the WT maximum value at about 5.75 Å$^{-1}$ for Zn/CS closely matched that of ZnPc, indicating the existence of dominant Zn–N bonds[29,40].

## Catalytic activity, reusability, and substrate applicability of Zn/CS in N-alkylation of nitroarenes with alcohols

To evaluate the catalytic performance of the Zn/CS catalyst, we applied it to the synthesis of N-alkylation compounds by employing the reaction of nitrobenzene with benzyl alcohol as the model. The reaction conditions were initially optimized by investigating various factors, such as solvent, type of base, reaction time, temperature, and catalyst dosage. As detailed in Table S3, the Zn/CS catalyst demonstrated superior catalytic performance in some nonpolar solvents during the screening of some common polar and nonpolar solvents (entries 1–11).

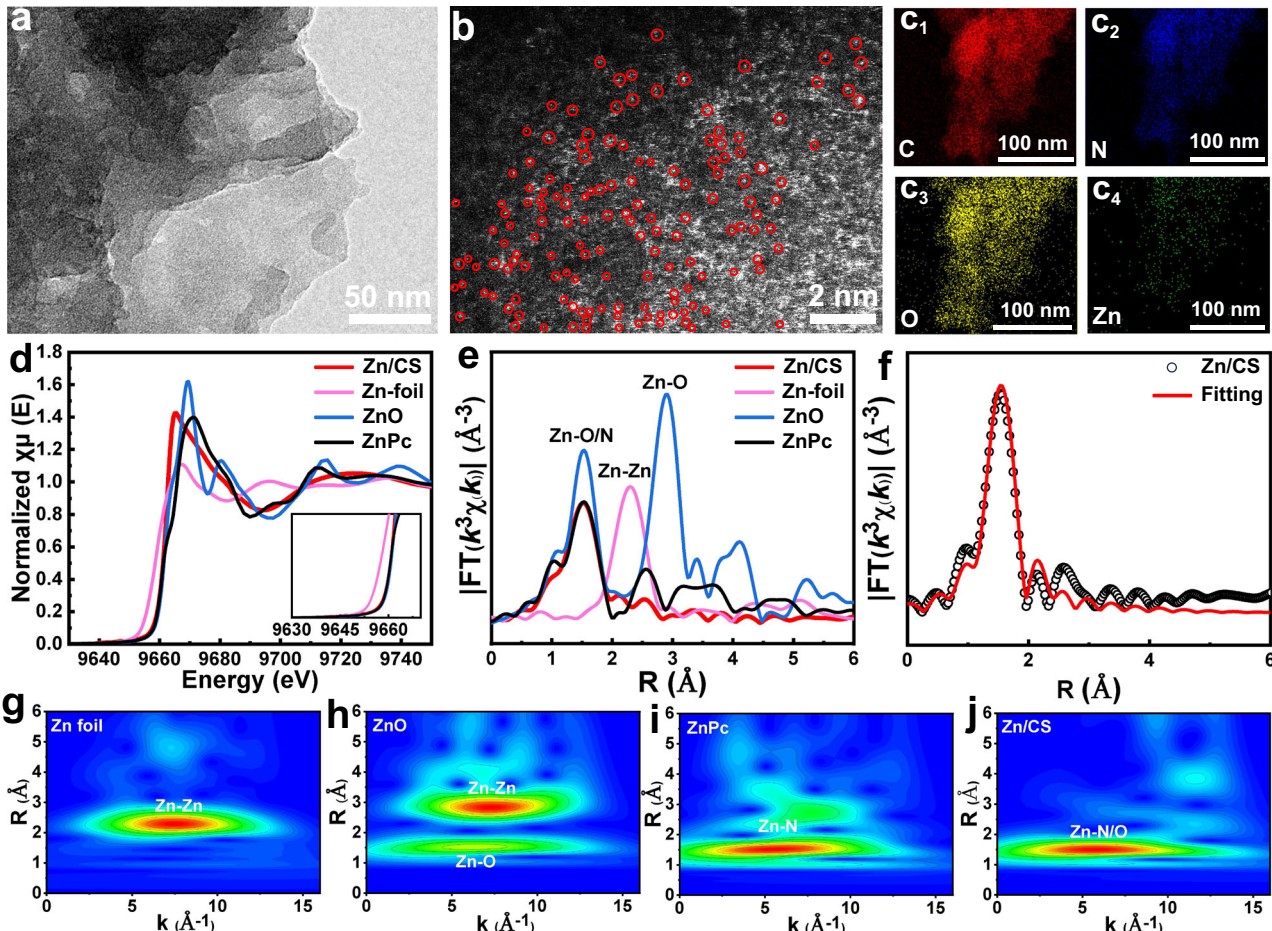

**Fig. 2 | TEM, HAADF-STEM, and XAS analysis of the Zn/CS catalyst. a** TEM image of the Zn/CS catalyst. **b** HAADF-STEM image of the Zn/CS catalyst, Zn atoms were marked with red circles. **c** Elemental mappings of HAADF-STEM for the Zn/CS catalyst. **d** XANES spectra, and **e** EXAFS spectra of the Zn foil, ZnO, ZnPc, and Zn/CS catalysts at the Zn K-edge. **f** Zn K-edge EXAFS (points) and curve fit (line) for the Zn/CS catalyst, shown in R-space. **g–j** Wavelet transform contour plots of the Zn foil, ZnO, ZnPc, and Zn/CS catalysts.

Specifically, a 95% yield of the desired product was achieved at 120 °C in petroleum ether over 24 h (Table S3, entry 11). Next, the effects of reaction temperature (Table S4) and time on catalytic activity were examined. As shown in Table S3, increasing the reaction time from 15 h to 24 h, 98% yield of N-benzylaniline in 21 h at 120 °C was detected (Table S3, entries 19–21). Regarding the reaction temperature, the yield of N-benzylaniline increased as the temperature raised from 90 to 120 °C, plateauing at 92% when the temperature reached 130 °C (Table S4). Furthermore, the type and amount of base were explored, which revealed that in various organic or inorganic bases reported in literatures, 98% yield of N-benzylaniline was achieved when the base was KOH with a quantity of 0.54 mmol after 21 h (Table S3, entry 21; Table S5). Notably, the Zn/CS could not achieve the deprotonation and the subsequent conversion of benzyl alcohol in the absence of KOH (Figs. S12, S13, and Table S5, entry 5). In the screening of catalyst amount, as the catalyst amount increased from 0.03 mol% ([Zn]: nitrobenzene, mol%) to 0.12 mol%, the yield of N-benzylaniline initially increased from 28% to 95%, but a further increase in catalyst amount led to a decrease in yield (Table S6). Additionally, the ratio of nitrobenzene to benzyl alcohol was also optimized, revealing that the desired yield of N-benzylaniline reached 98% at a ratio of 1:3 (Table S7).

To further evaluate the catalytic performance of the Zn/CS catalyst, we compared it with commercial Zn/C, nano-Zn, homogeneous $Zn(NO_3)_2 \cdot 6H_2O$, and blank chitosan materials. The Zn content was standardized to 0.09 mol% ([Zn]: $PhNO_2$), and the samples were taken at different time intervals to monitor the formation of target products. As shown in Figs. 3a and S14, the Zn/CS catalyst exhibited the highest catalytic activity among these tested catalysts. The commercial Zn/C catalyst could only achieve an 8% yield, emphasizing the advantages of using chitosan as the carrier. Although the homogenous $Zn(NO_3)_2 \cdot 6H_2O$ catalyst yielded 16% after 21 h, it was not recyclable. The commercial nano-Zn could yield only 8% after 21 h, further emphasizing the importance of the natural skeleton of chitosan. In comparison, the blank chitosan carrier without Zn did not facilitate the synthesis of target substance, confirming the essential role of Zn species in the catalytic process. Notably, as shown in Fig. 3b and Table S8, the synthesized Zn/CS catalyst exhibited excellent catalytic activity compared to various reported heterogeneous catalysts, with the highest TOF and broader substrate scopes in lower reaction temperature and catalyst usage, demonstrating the successful synthesis of N-alkylation compounds in a green and efficient manner. In all, the Zn/CS catalyst exhibited superior catalytic activity, which may be attributed to the following reasons: (1) the abundant hydroxyl/amino groups and nanoporous structure in natural skeleton of chitosan, which provide numerous binding sites for Zn atoms, as well as facilitated the exchange and diffusion of substrates; (2) the high-electronegativity N/O elements in chitosan skeleton act as ligands to significantly modulate the coordination environment and electronic structure of Zn atoms, enhancing their catalytic performance.

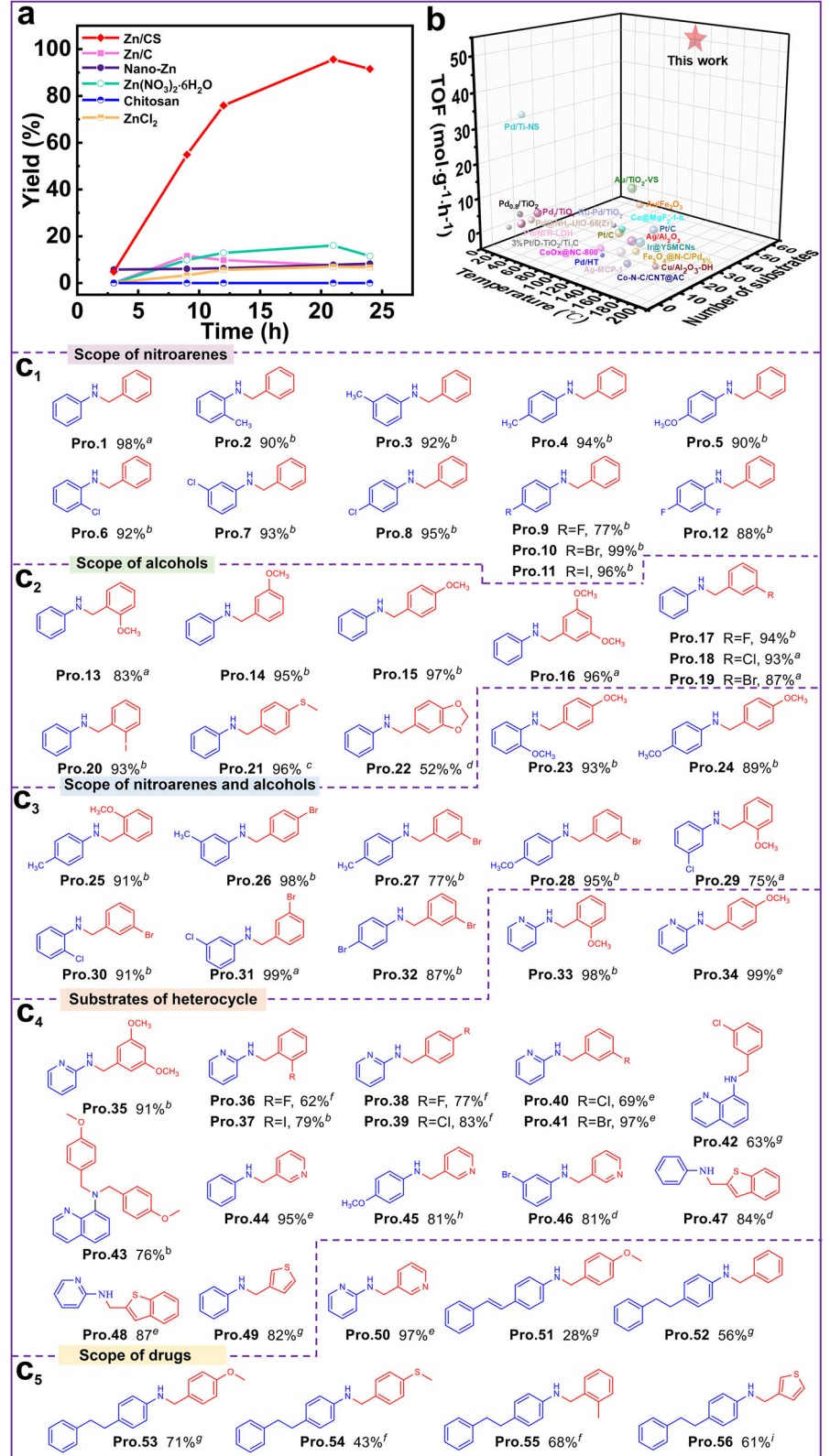

**Fig. 3 | Catalytic activity and substrate scope. a** Kinetic curves and **b** comparison of TOFs for various commercial and reported catalysts in the model reaction of nitrobenzene and benzyl alcohol. **c₁–c₅** Substrate applicability: [a] Reaction conditions: nitro compounds (0.2 mmol), alcohols (0.6 mmol), KOH (0.54 mmol), petroleum ether (6 mL), Zn/CS (0.09 mol% [Zn], Zn: PhNO₂), reacted at 120 °C for 21 h. [b] Reaction time was 36 h. [c] Toluene (4 mL), Zn/CS (0.17 mol% [Zn]), reacted at 130 °C for 24 h. [d] Zn/CS (0.17 mol% [Zn]), reacted at 130 °C for 36 h. [e] Reacted at 120 °C for 48 h. [f] Zn/CS (0.17 mol% [Zn]), reacted at 130 °C for 48 h. [g] Toluene (4 mL), Zn/CS (0.17 mol% [Zn]), reacted at 130 °C for 36 h. [h] Toluene (4 mL), Zn/CS (0.17 mol% [Zn]), reacted at 130 °C for 48 h. [i] Zn/CS (0.3 mol% [Zn]), reacted at 130 °C for 21 h. Yields of the product were isolated yield, and the corresponding NMR spectra were shown in the supporting information.

We further investigated the reusability of the Zn/CS catalyst by performing a template reaction with nitrobenzene and benzyl alcohol under the optimal conditions. As shown in Fig. S15, the yield of the target product remained at 91% after 5 cycles, indicating minimal loss of catalytic activity. ICP analysis of the catalyst after 5 cycles revealed a slight decrease in Zn loading from 1.110 to 1.083%, suggesting that only a small amount of Zn leached into the reaction solution (Fig. S16). TEM images of the catalyst after 5 cycles demonstrated that the Zn/CS catalyst still exhibited well-dispersed Zn single atoms, with no aggregation (Fig. S17). FT-IR analysis indicated almost no structural changes after 5 cycles (Fig. S18), confirming the good stability of the chitosan carrier. XRD analysis in Fig. S19 displayed characteristic peaks corresponding to chitosan crystal planes both before and after cycling, with no observable diffraction peaks attributed to Zn, further demonstrating the stability. XPS spectrum of Zn $2p$ displayed two peaks at 1021.78 and 1044.89 eV after 5 cycles, which were similar to those of the fresh Zn/CS with $Zn^{+\delta}$ state (Fig. S20). In conclusion, Zn single atoms were tightly anchored on chitosan support and uniformly dispersed, providing the catalyst with excellent reusability. The key factors contributing to this stability were as follows: firstly, the chitosan skeleton contained abundant functional groups ($-NH_2$, $-OH$, etc.) that coordinated well with Zn atoms, thereby stabilizing Zn tightly; secondly, the Zn/CS catalyst possessed nanoporous structure and large surface area, providing rich attachment sites for Zn atoms; thirdly, the electron flow between chitosan support and Zn atoms proved by XPS could enhance the interaction between chitosan and Zn.

Under the optimal reaction conditions, the general applicability of the Zn/CS catalyst was evaluated. As shown in Fig. 3c_1–c_2, when the derivatives of nitrobenzene or benzyl alcohol were used as substrates, whether the nitro-compounds or alcohols were substituted with electron-withdrawing or electron-donating groups, which could successfully yield the corresponding desired products with yields in the range of 52–99%. Furthermore, the yields of the desired products were influenced by the positions of the substituents, such as the nitro-compounds or alcohols substituted at the ortho position exhibited lower yields compared to those substituted at the meta or para positions (for examples, Pro.2–Pro.4, Pro.6–Pro.8, and Pro.13–Pro.15), which may be attributed to the steric effects of the substituents[41]. Notably, when nitro-compounds or alcohols were substituted with halogen atoms, the corresponding halogenated compounds were successfully achieved, with no significant dehalogenation (for examples, Pro.6–Pro.12, Pro.17–Pro.20). Interestingly, some disubstituted derivatives of nitrobenzene or benzyl alcohol with greater steric hindrance also exhibited good yields of the target products, ranging from 52 to 96% (for examples, Pro.12, Pro.16, and Pro.22). Additionally, we also investigated the catalyst's applicability when both nitro-compounds and alcohols were substituted. Encouragingly, when both nitro-compounds and alcohols were substituted with different groups, the corresponding target products were obtained with yields ranging from 75 to 99% (Fig. 3c_3, Pro.23–Pro.32).

Subsequently, the effects of nitro-compounds or alcohols substituted with heterocycles were investigated, such as pyridine, thiophene, quinoline, etc. As shown in Fig. 3c_4, the Zn/CS catalyst effectively catalyzed the borrowing hydrogen reactions of these heterocyclic amines or alcohols, achieving satisfactory product yields of 62–99% (Pro.33–Pro.49). Impressively, several important pharmacophores and biologically relevant compounds were successfully synthesized (Fig. 3c_5), including a pharmacophore from anticancer drugs approved by the food and drug administration between 2015 and 2020, with yield of 97% (Pro.50)[42]. Using the Zn/CS catalyst, (E)−1-nitro-4-vinylbenzene was smoothly converted into drug candidates for Alzheimer's treatment (Pro.51), along with a series of its derivatives (Pro.52–Pro.56)[43,44].

## Study on the reaction pathway/mechanism

To elucidate the reaction pathways/mechanisms of the borrowing hydrogen reaction between nitroarenes and alcohols, nuclear magnetic resonance (NMR) spectroscopy, gas chromatography (GC) and deuterium labeling experiments were conducted to monitor the reaction mixture at different time intervals. Typically, the above borrowing hydrogen reaction proceeded through the following pathways: (1) the alcohol was oxidized to an aldehyde by the catalyst, generating the metal-H species; (2) the formed metal-H reduced the nitroarene to an arylamine; (3) the aldehyde coupled with the arylamine to form an imine intermediate; (4) the imine intermediate was further reduced by the metal-H to yield N-benzylaniline. To confirm these pathways, kinetic monitoring of the model reaction between nitrobenzene and benzyl alcohol was performed at 0.5, 3, 6, 12, and 21 h by NMR and GC. As shown in Fig. 4a, b, the substrate of benzyl alcohol gradually decreased as the reaction progressed, and a peak corresponding to benzaldehyde was detected, indicating that benzyl alcohol was successfully oxidized to benzaldehyde. In the GC kinetic profile, during the first 30 min, nitrobenzene was observed to convert into intermediates, such as nitrosobenzene, azobenzene, aniline, and the amount of nitrobenzene steadily decreased. Similarly, in the NMR kinetic profile, peaks corresponding to the intermediate N-phenylhydroxylamine and aniline were also observed (the peak of the $-NH_2$ group in aniline may have overlapped with the $-NH-$ peak in the target product of N-benzylaniline, as shown in Fig. S21), further confirming the conversion of nitrobenzene to aniline. Moreover, in the kinetic profiles of both NMR and GC, it was found that the amount of the imine intermediate, formed from the coupling of aniline and benzaldehyde, initially increased and then decreased, while the amount of the target product N-benzylaniline gradually increased. Notably, the $^1H$ NMR and GC kinetic spectra demonstrated the substantial amounts of azobenzene as the reaction progressed compared to the trace phenylhydroxylamine intermediate, indicating that the path of azobenzene may be the dominant reaction pathway. These results elucidated the borrowing hydrogen process, which proceeded through four key stages: alcohol oxidation, nitroarene reduction, aldehyde-amine coupling, and imine reduction.

To further validate the reaction pathways and borrowing hydrogen process described above, a series of parallel experiments were conducted. As shown in Fig. 4c, benzyl alcohol alone was reacted in the presence of Zn/CS, resulting in its oxidation to benzaldehyde (step 1). The reaction of benzyl alcohol with benzaldehyde led to an accumulation of benzaldehyde, further indicating the existence of the borrowing hydrogen process (step 2). Subsequently, nitrobenzene and aniline were each reacted separately with benzaldehyde, and it was found that no reaction occurred between nitrobenzene and benzaldehyde after 21 h (step 3). However, aniline and benzaldehyde underwent a coupling reaction to produce only the intermediate N-benzylideneaniline (step 4), demonstrating that there was no "hydrogen donors" in the absence of benzyl alcohol to generate Zn−H, thus achieving no conversion of N-benzylideneaniline to N-benzylaniline by Zn−H. As a control, the reactions of benzyl alcohol with nitrobenzene and benzyl alcohol with aniline (steps 5–6) were conducted, which both successfully yielded the target products with good yields after 21 h. Furthermore, in the presence of Zn/CS catalyst, the intermediate N-benzylideneaniline was reacted separately with benzaldehyde and benzyl alcohol. It indicated that no new substances were produced in the reaction of N-benzylideneaniline and benzaldehyde (step 7). However, in the reaction of N-benzylideneaniline and benzyl alcohol, the target product N-benzylaniline was produced with a yield of 98% (step 8), further demonstrating that the reduction of N-benzylideneaniline to N-benzylaniline occurred through the hydrogen donors of Zn−H generated by benzyl alcohol. Notably, the reaction of the intermediate N-benzylideneaniline with benzyl alcohol did not yield the N-benzylaniline in the absence of Zn/CS catalyst, indicating

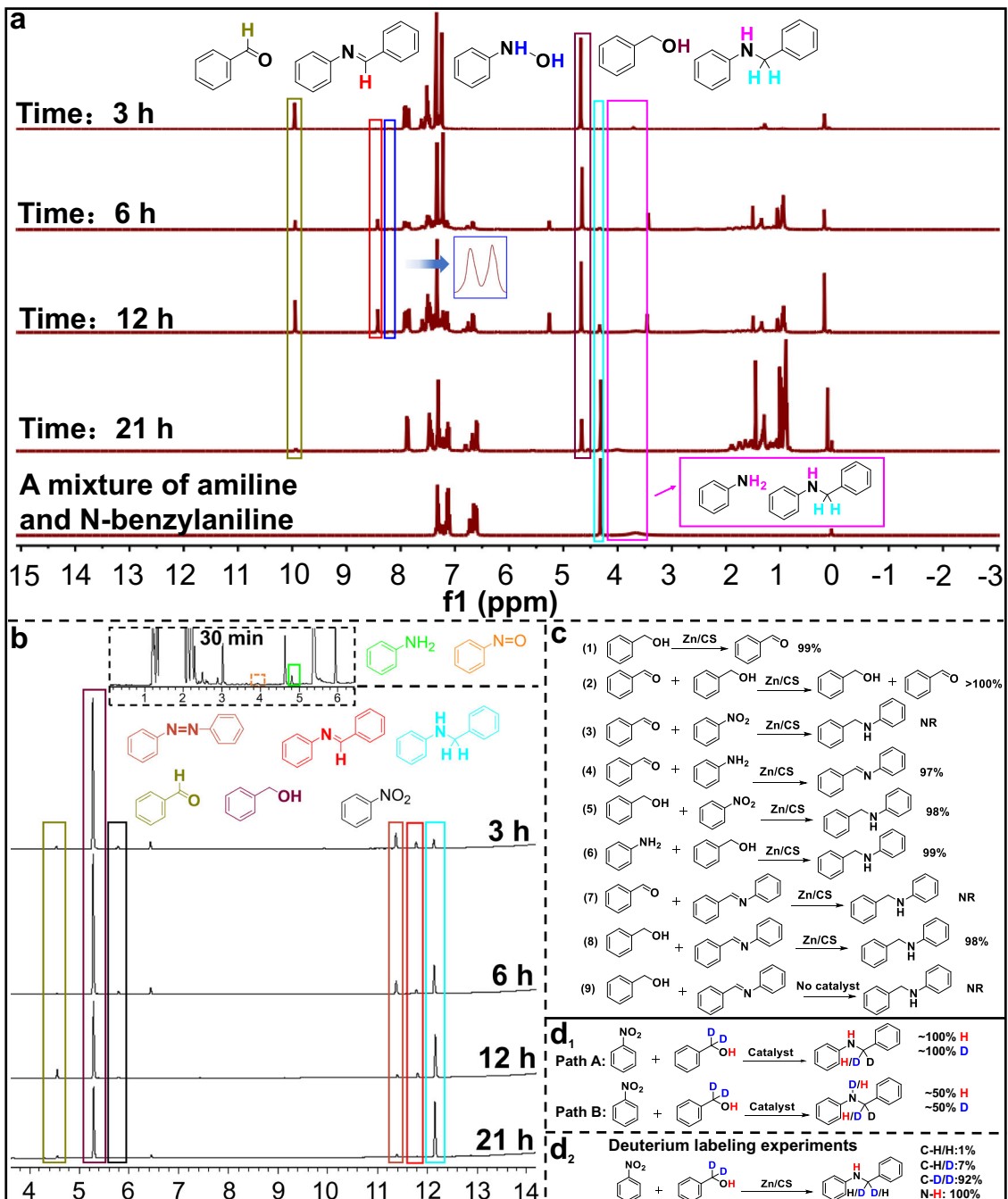

**Fig. 4 | The reaction pathway/mechanism. a** NMR spectra over time, and **b** GC spectra over time for the borrowing hydrogen reaction of nitrobenzene and benzyl alcohol catalyzed by Zn/CS. **c** The parallel experiments catalyzed by Zn/CS. **d** Two different pathways and deuterium labeling experiments.

that benzyl alcohol alone could not initiate the borrowing hydrogen process and form Zn–H without the Zn/CS catalyst (step 9). These parallel experimental results were consistent with the above NMR and GC data, further confirming the borrowing hydrogen reaction process between nitrobenzene and benzyl alcohol.

To further investigate the mechanism of the above borrowing hydrogen reaction, deuterium labeling experiment was conducted by using α-C–H deuterated benzyl alcohol as the hydrogen source. According to the previous reports[45,46], the above borrowing hydrogen reaction catalyzed by transition metals mainly involved the metal hydride pathway[47,48]. Therefore, two commonly reported metal hydride reaction pathways were proposed (as presented in Fig. 4d1, paths A–B and Fig. S22). As shown in Figs. 4d2 and S23–26, ¹H NMR

analysis revealed the formation of three types of N-benzylaniline, with C–H/H (1%), C–H/D (7%) and C–D/D (92%), and the N–H bond in N-benzylaniline showed no deuterated H atoms. High resolution mass spectrometry (HR-MS) analysis further confirmed the presence of N-benzylaniline with types of C–H/H, C–H/D, and C–D/D (Fig. S25), particularly the predominant C–D/D type, and the N–H bond in N-benzylaniline had not been deuterated. These results indicated that our reaction mixture predominantly proceeded through the Zn–H pathway (path A).

Based on the above results, the proposed reaction mechanism was illustrated in Fig. S27. First, under the action of KOH, the H atom on the hydroxyl group of benzyl alcohol was removed, forming a phenylmethanolate (A), which rapidly coordinated with the Zn single

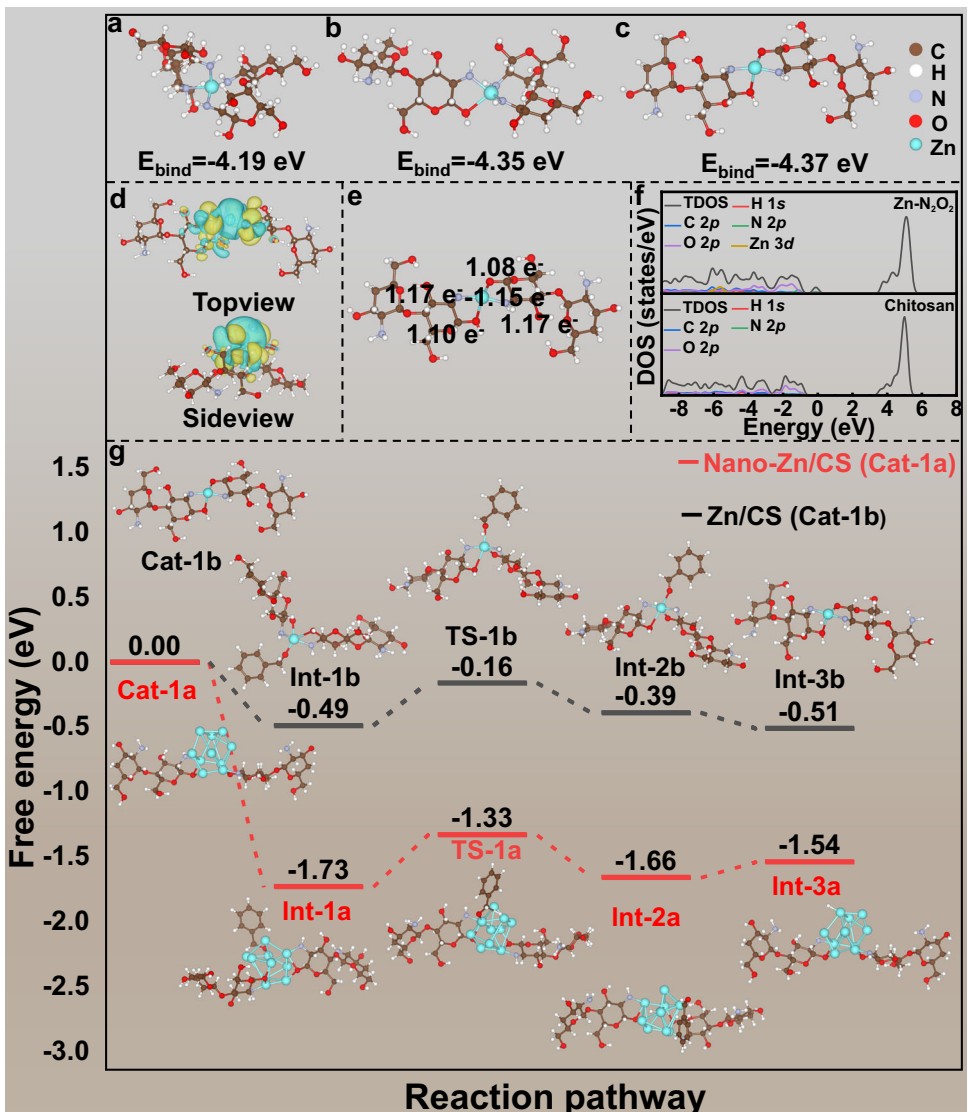

**Fig. 5 | DFT calculations.** The models and binding energies of **a** ZnN$_4$-CS, **b** ZnN$_3$O$_1$-CS, and **c** ZnN$_2$O$_2$-CS. **d** Top and side views of the charge density difference of ZnN$_2$O$_2$-CS, with yellow and blue regions indicating charge accumulation and depletion, respectively. **e** Bader charge analysis of the ZnN$_2$O$_2$-CS. **f** The density of states for the blank chitosan and ZnN$_2$O$_2$. **g** The reaction pathway and corresponding Gibbs free energy profiles for benzyl alcohol oxidation by using the Zn/CS and nano-Zn/CS catalysts. Figures **d**–**f** were obtained using VASP, while the other panels were generated using DMol³.

atoms in Zn/CS to produce intermediate Int-1b, followed by dehydrogenation to produce benzaldehyde (B) and Zn–H. Meanwhile, nitrobenzene was reduced by Zn–H to intermediates, such as nitrosobenzene, N-phenylhydroxylamine, azobenzene, and ultimately to the target aniline. Subsequently, the former benzaldehyde and aniline underwent a coupling reaction, releasing one molecule of H$_2$O to generate the imine intermediate (E). Finally, the imine intermediate was reduced by Zn–H to form the target product N-benzylaniline (F), regenerating the Zn/CS catalyst and completing the catalytic cycle.

## DFT calculations
Based on the XPS and EXAFS results, the Zn single atoms in chitosan support exhibited a Zn–N/O structure with coordination number of 4, and Zn–N coordination was dominant. Therefore, we attempted to establish the coordination structure models of ZnN$_4$-CS, ZnN$_3$O-CS, and ZnN$_2$O$_2$-CS. As shown in Fig. 5a–c, DFT was employed to optimize the structures and calculate the binding energies of ZnN$_4$-CS, ZnN$_3$O-CS, and ZnN$_2$O$_2$-CS, which indicated that the binding energies the above coordination models were −4.19, −4.35, and −4.37 eV,

respectively (Fig. S29). Moreover, during the calculation, it was found that compared to other coordination models, the active Zn sites in the ZnN$_2$O$_2$-CS model were more readily exposed, thereby reducing the steric hindrance between the reactants and Zn atoms, which was in line with the superior catalytic performance of the Zn/CS catalyst. Therefore, the ZnN$_2$O$_2$-CS model was chosen as the best coordination structure.

To better understand the electron transfer and distribution between the Zn single atoms and the chitosan support, charge density differential ($\Delta\rho$) and Bader charge analyses were performed on the optimal coordination model of ZnN$_2$O$_2$-CS. As shown in Fig. 5d, some electrons were transferred from the Zn single atoms and accumulated on the N/O atoms of chitosan (the yellow regions indicated electron accumulation, while the blue regions represented electron depletion). The Bader charge analysis revealed that the Zn atoms donated 1.15 e⁻ to the neighboring N/O atoms, while the N atoms and O atoms accepted some electrons, resulting in an asymmetric ZnN$_2$O$_2$-CS electronic structure (Fig. 5e). This asymmetric electronic distribution was more easily polarized, which could facilitate the adsorption or activation

between the catalyst and the reaction substrates, so as to promote the catalytic reaction. We believed that the Zn atoms with a positive charge after losing electrons could promote the adsorption with phenylmethanolate (Ph-CH$_2$O$^-$), triggering the dehydrogenation of alcohols and facilitating the whole reaction. Therefore, this electron-deficient and asymmetric Zn-N$_2$O$_2$ coordination sites may be the fundamental reason for the good catalytic activity of Zn/CS. Furthermore, the density of states (DOS) of Zn/CS was analyzed. As shown in Figs. 5f and S30, the incorporation of Zn atoms introduced new electronic states near the Fermi level, indicating a significant modification of the local electronic structure of the chitosan support. The partial DOS revealed that these new states involved Zn 3$d$ and the neighboring N/O atoms' orbitals, pointing to strong orbital hybridization between Zn and the N/O atoms in chitosan, confirming the successful coordination of Zn. Consequently, the electronic conductivity of the catalyst could be enhanced. The narrowed bandgap and increased DOS near the Fermi level indicated the improved charge-carrier mobility. The modification of the electronic structure was expected to facilitate the electron transfer during catalytic reactions, thereby enhancing the catalyst's activity. The above analyses indicated that after the incorporation of Zn single atoms, a significant electron rearrangement occurred between Zn and N/O atoms in chitosan, forming an asymmetric ZnN$_2$O$_2$-CS structure with electron-deficient character.

To validate the role of the asymmetric Zn-N$_2$O$_2$ structure on catalytic activity, a control experiment was conducted by using nano-Zn/CS (the model was shown in Fig. S31; TEM image and preparation method was shown in Fig. S32). The variation of Gibbs free energy in this borrowing hydrogen reaction was studied by using DFT simulation. Initially, a kinetic study was performed on the model reaction to determine the RDS. As illustrated in Fig. S33, in the reactions involving nitrobenzene with benzyl alcohol, as well as N-benzylideneaniline with benzyl alcohol, both the reactions exhibited a positive association, indicating the first-order kinetics. Notably, compared to the conversion of nitrobenzene (represented the reaction of nitrobenzene to aniline) or the conversion of N-benzylideneaniline (represented the reaction of imine intermediate to N-benzylaniline), the rate constant ($K = 0.087$) for the conversion of benzyl alcohol (represented the reaction of benzyl alcohol to benzaldehyde) was the smallest, suggesting that the dehydrogenation of benzyl alcohol constituted the RDS in the borrowing hydrogen reaction between nitrobenzene and benzyl alcohol.

Herein, the Gibbs free energy changes involved in the RDS of dehydrogenation of benzyl alcohol to benzaldehyde were investigated by using the Zn/CS catalyst as an example. As shown in Fig. 5g, the process commenced with the deprotonation of benzyl alcohol in the presence of KOH, followed by the adsorption of the oxygen atom in Ph-CH$_2$O$^-$ onto the Zn atom, yielding an intermediate (Int-1b). Subsequently, the H atom on the methylene group of Ph-CH$_2$O$^-$ interacted with the Zn atom, forming a transition state of Ph-CH$_2$O* (TS-1b). Finally, the H atom on the methylene group of TS-1b was transferred to the Zn atom, resulting in an intermediate Int-2b, ultimately producing benzaldehyde and Zn−H (Int-3b). Notably, in the process of forming transition state TS-1b, the Zn/CS catalyst required overcoming an energy barrier of 0.33 eV, which was lower than 0.40 eV barrier associated with Zn$_{11}$N$_2$O$_2$-CS, indicating that the Zn/CS catalyst could significantly reduce the reaction barrier for the dehydrogenation conversion of benzyl alcohol. Collectively, these data demonstrated that the Zn/CS catalyst was more favorable for the oxidation of benzyl alcohol during the borrowing hydrogen reaction. The essential reason for the high catalytic performance of Zn/CS catalyst may be due to that the asymmetric Zn-N$_2$O$_2$ sites with electron-deficient character had a stronger adsorption energy for Ph-CH$_2$O$^-$. Thus, the Zn/CS was more likely to form the Zn−H and Zn−O bonds with Ph-CH$_2$O$^-$ to generate the transition state TS-1b (Ph-CH$_2$O*), thus facilitating the dehydrogenation of benzyl alcohol and driving the whole reaction. Meanwhile, the

changes of Gibbs free energies for the hydrogenation of nitrobenzene, and the hydrogenation of imine were also provided, as presented in Figs. S28 and S34. It could be seen that the energy barrier required to form the transition states during the hydrogenation of nitrobenzene or imine was lower than that required to form the transition state TS-1b during the oxidation of benzyl alcohol. These data further proved the dehydrogenation of benzyl alcohol to benzaldehyde as the RDS.

## Discussion

In conclusion, for the first time, a novel bio-inspired Zn SACs with asymmetric Zn-N$_2$O$_2$ sites via the natural skeleton of biomass chitosan was successfully synthesized. Through the sol-gel strategy, the support of chitosan microspheres exhibited a nanoporous structure and large surface area, which facilitated the attachment of Zn atoms and the exchange of reactants. Additionally, the abundant N/O functional groups on chitosan can regulate the coordination environment and electronic structure of Zn atoms like ligands, as well as effectively anchor the Zn atoms. The synthesized catalyst was applied in the borrowing hydrogen reaction between nitroarenes and alcohols, which demonstrated excellent catalytic activity compared to range of commercial and previously reported catalysts based on the asymmetric Zn-N$_2$O$_2$ coordination structure induced by the chitosan skeleton, with the highest TOF of 51.85 h$^{-1}$. Meanwhile, the Zn/CS catalyst also displayed broad substrate applicability (56 examples) and outstanding recycling stability, indicating its potential for industrial applications. The corresponding catalytic pathway/mechanism was also elucidated, which revealed that the asymmetric Zn-N$_2$O$_2$ sites with electron-deficient character were more conducive to formation of Zn−H and Zn−O bonds in transition state Ph-CH$_2$O* to trigger the dehydrogenation of alcohols, thus facilitating the whole reaction. This study represents the first use of a biomass-derived Zn SACs in the borrowing hydrogen reaction between nitroarenes and alcohols, which provides a new approach for the efficient synthesis of N-alkylation compounds and contributes to the development and utilization of biomass resources.

## Methods
### Preparation of the chitosan microspheres
Chitosan microspheres were prepared using the sol-gel method. Specifically, 4.5 g of chitosan powder was dissolved in 100 g of solution consisting of 4.32 wt% LiOH/6.71 wt% KOH/7.68 wt% urea/77.28 wt% H$_2$O. The resulting suspension was frozen at −35 to −40 °C for 2 h, then thawed at room temperature with stirring. This freeze-thaw cycle was repeated three times. Subsequently, 10 g of Span 85 and 175 g of isooctane were added to a 1 L three-necked flask and stirred in an ice-water bath for 30 min. The prepared chitosan solution was then added and stirred for an additional 1 h before removing the ice bath. The mixture was stirred at approximately 60 °C for 30 min to form a chitosan microsphere emulsion, which was further transferred to a mixed solution of ethanol and H$_2$O (9:1, v/v) and stirred for 1 h. Finally, the obtained microspheres were filtered, washed repeatedly with ethanol and deionized H$_2$O, and then freeze-dried for subsequent use.

### Preparation of the Zn/CS catalyst
By employing the impregnation method, 70 mg of zinc nitrate hexahydrate (Zn(NO$_3$)$_2$·6H$_2$O) was dissolved in 5 mL of H$_2$O. Subsequently, 500 mg of the previously prepared chitosan microspheres were dispersed in 200 mL of H$_2$O and soaked for 15 min. The Zn(NO$_3$)$_2$·6H$_2$O solution was then added dropwise to the chitosan microsphere suspension, which was stirred in an ice bath for 1 h, followed by continued stirring at room temperature for 3 h. The resulting suspension was filtered and dried to obtain the precursor Zn$^{2+}$/CS catalyst. Finally, the precursor Zn$^{2+}$/CS catalyst was activated at 250 °C (with a heating rate of 2 °C/min) in an argon (Ar) atmosphere for 2 h to produce the Zn/CS catalyst.

**Borrowing hydrogen reaction of nitroarenes with alcohols**

The catalytic performance of the Zn/CS catalyst was evaluated through the borrowing hydrogen reaction between nitroarenes with alcohols. In the model reaction, 0.2 mmol of nitrobenzene, 0.6 mmol of benzyl alcohol, 0.54 mmol of KOH, 0.09 mol% [Zn] ([Zn]: $PhNO_2$, mol%), and 6 mL of petroleum ether were added to a reaction flask equipped with a magnetic stirrer. The reaction was conducted at 120 °C for 21 h, and the yield of the product was determined by GC upon completion.

For substrate extension studies, reactions were carried out by introducing 0.2 mmol nitroarenes, 0.6 mol alcohols, 6 mL solvent of petroleum ether, 0.54 mmol KOH, and 0.09 mol% [Zn] ([Zn]: $PhNO_2$, mol%). These reactions were conducted for 21–48 h at temperatures between 120 and 140 °C. After the reaction, the reaction products were purified using column chromatography with a petroleum ether/ethyl acetate mixture as the eluent, and the isolated products were characterized by NMR. Additionally, catalyst recycling experiments were conducted using nitrobenzene and benzyl alcohol as the model reaction, and the reaction conditions were the same as above.

## Characterization and computational details

To determine the structure of the materials and the conversion of each substrate in the reaction, various physicochemical characterizations were performed, and the details were shown in Supplementary Information. The computational details were shown in Supplementary information (part of S2. Computational details).

## Data availability

The data supporting the findings of this study are available within the paper, the Supplementary Data 1, and the Supplementary Information. Source data are provided in this paper. All data are available from the corresponding author upon request. Source data are provided with this paper.

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

## Acknowledgements

This work was supported by the Key Project of Natural Science Foundation of Guizhou Province (no. ZK[2023]Key 025, X.P.), the National Natural Science Foundation of China (no. 52063008, X.P.), the Lightweight Materials Engineering Research Center of the Education Department of Guizhou (no. [2022] 045, X.P.), the Guizhou Province Science and Technology Project (no. [2024] Normal 072, X.P.), and the Qiannan Normal College for Nationalities University Science Park (No. ZZSG[2024] 002, X.P.). Special thanks to the Science and Technology Innovation Platform of Guizhou Normal University for its assistance in our research work.

## Author contributions

Y.H. wrote-original draft, conceptualization, investigation, data curation, and methodology. Y.L. performed the formal research. Q.Z. conceived and designed the investigation. M.H. contributed to the discussion. X.C. analyzed the data, X.L., W.L., and X.Y. assisted with interpretation of results and provided valuable advice on data analysis. A.L., X.P. supervised and guided the project and revised the manuscript. All authors discussed the results and commented on the manuscript.

## Competing interests

The authors declare no competing interests.

## Additional information

**Supplementary information** The online version contains Supplementary material available at https://doi.org/10.1038/s41467-026-70172-1.

