## [Transparent Peer Review File · Nature Communications]

Bio-inspired Asymmetric Zn-N₂O₂ Single-atom Catalysts via Natural Skeleton for Efficient N-alkylation of Nitroarenes with Alcohols

Corresponding Author: Professor Aiwen Lei

Version 0:

Reviewer comments:

Reviewer #1

(Remarks to the Author)

This manuscript reported for the first time a novel bio-inspired Zn SACs with asymmetric Zn-N₂O₂ sites by utilizing biomass chitosan (Zn/CS), and achieved the first instance of heterogeneous Zn SACs in borrowing hydrogen reaction between nitroarenes and alcohols. Notably, the Zn/CS catalyst exhibited the highest TOF among various heterogeneous catalysts, as well as broad substrate scopes and excellent reusability, and the catalytic mechanism was revealed clearly. This work has significant original innovation, and is well written. It will guide researchers to find more interesting and efficient catalysts for the N-alkylation reaction. Therefore, I recommend this study for publication in Nature Communications with minor revision. To improve this work, I have the following comments and suggestions:

- (1) Why was chitosan selected from various available biomasses for this study? The formation mechanism of chitosan microspheres should be elucidated.
- (2) The Zn/CS catalyst was activated at 250 °C in an argon atmosphere from the Zn²⁺/CS precursor. Why was the Zn²⁺/CS precursor catalyst not used directly for catalysis?
- (3) In this study, petroleum ether was used as the reaction solvent. Had the authors tried using mixed solvents? A large number of literatures suggested that mixed solvents could have an unexpected promoting effect in organic reactions.
- (4) Please provide data on nano-Zn/chitosan catalyst for the catalysis of benzyl alcohol and nitrobenzene, and compare it with the catalytic activity of the synthetic Zn SACs.
- (5) The authors stated that the yield of N, N-benzylaniline decreased when increasing the catalyst amount. But there was no evidence, it should be provided.
- (6) The substrates of aromatic alcohols were studied. What about the aliphatic alcohols with chain or ring structures in the substrate expansion? Such attempts should be included in the manuscript.

Reviewer #2

(Remarks to the Author)

In this work, a Zn single-atom catalyst based on chitosan support is used for the N-alkylation of nitrobenzenes with alcohols showing high activity and broad substrate scope under mild conditions, which is uncommon using non-noble metal SACs. The work also includes a mechanistic study that agrees with a hydrogen borrowing reaction (several steps have been isolated). DFT calculations are also included for one reaction step. While the experimental results are sound, the computational study and mechanistic proposal have several inconsistencies that require major revisions (see details). The computational Figure is also hard to read.

- 1) The initial models seem neutral, which could correspond to a Zn²⁺ neutralized by deprotonated NH₂ or OH groups after reaction with the base. However, this is not mentioned in the text, and it is very difficult to see in Figure 5 the presence of OH or O groups. In addition, different models can be constructed by changing the deprotonated groups. Has this been considered? How does the selected model fit the experimental data? eg EXAFS results in Table S2 (e.g., Zn-N/O distance 1.99 Å) or the WT maximum value indicating the existence of dominant Zn-N bonds.
- 2) If the support is deprotonated, the statement on page 15 mentioning "the NH₂ and OH groups from chitosan is the reason for the catalyst stability" should be corrected.
- 3) The same section mentions that the electron flow between chitosan and Zn atoms enhances the interaction between Zn

- and the support. This suggests that the support reduces the Zn atoms. Is there any support for this statement?
- 4) For the Bader analysis and the computed energy pathway, it would be helpful to compare the results with a similar model, which is not reactive (e.g., Zn(NO₃)₂), instead of a Zn nanoparticle, which is more challenging to model. Is there any support for the size and shape of the nanoparticle used in the calculations?
 - 5) The energy profile assumes that the base reacts with the alcohol substrate instead of the support. Could the support deprotonate the alcohol instead of the base?
 - 6) Another way to test the model used for the reaction would be to compute the hydrogenation of imine and show that it has a lower energy barrier than the dehydrogenation of the alcohol, as suggested experimentally.
 - 7) The discussion on deuterium labeling on page 19 is unclear. The same is true for the equations in Figure 4 d1 and d2. This should be clarified in the text and with some more explicit schemes, either in the manuscript or in the supporting information.
 - 8) On page 23, it is unclear what type of kinetic study has been done to conserve the computational resources. Figure S26 is not given in the supporting information.
 - 9) On the same page, the sentence "The Gibbs free energy changes involved in this borrowing hydrogen reaction were investigated in combination with DFT simulations" is unclear.
 - 10) It is recommended that all computed structures in the manuscript be included in the supporting information as an xyz file to facilitate their visualization using different software.

Reviewer #3

(Remarks to the Author)

Reviewer #4

(Remarks to the Author)

In this manuscript, Huang et al. reported the application of bio-inspired Zn single-atom catalysts (SACs) with asymmetric Zn-N₂O₂ sites for the borrowing hydrogen reaction between nitroarenes and alcohols. They found that the chitosan-derived support enhances Zn atom dispersion and electronic modulation, leading to exceptional catalytic activity with the highest turnover frequency among reported heterogeneous catalysts. Their DFT computations helped gain deeper insights. In general, this work presents a new strategy for biomass-derived non-noble metal SACs, offering a highly efficient, green, and recyclable catalyst for N-alkylation reactions under mild conditions. I am counting on the experimental peers to evaluate the experimental sections. Below I am only commenting on the theoretical part.

The authors performed DFT computations to help reveal the underlying mechanism, specifically, they revealed that the electron-deficient Zn-N₂O₂ sites can facilitate the formation of key Zn-H and Zn-O intermediates, thus accelerating alcohol dehydrogenation, and Zn atoms serve as active centers for charge redistribution, enhancing catalytic efficiency. Generally, the computations helped understand/validate the experimental observations, and provide a predictive framework for designing more efficient single-atom catalysts for borrowing hydrogen reactions. However, more careful work should be done about the presentation and description of the computational methods and results.

1. In the main text, it states "All density functional theory (DFT) calculations were performed using the 527 DMol3 package, Vienna Ab-initio Simulation Package (VASP), and Projector 528 augmented-wave method (DS-PAW)"

The authors should specify which software was used for which calculations, Or, if all calculations were done in one package, state explicitly which one.

2. Since the computational details are given in the Supporting Information, for the section, S2. Computational Details, the description is also not clear.

It seems that the complex was simulated by a molecular cluster model, most computations were done with Dmol3 code, including geometry optimization, energetic evaluations and reaction pathway search. VASP code was only used to get Bader charge, density of states, and charge density difference for the complex, and a large supercell, (30x30x30) angstrom, was used to simulate cluster models.

It seems that for Figure 5, the important figure about theoretical studies, d-f are by VASP, while others are by DMol3. If yes, clearly specify them in the figure caption and also in the main text.

In Figure 5f, both total density of states and partial density of states are presented, but the figure is too small. A larger figure can be given in the Supporting Information, and some more discussions are helpful.

3. For VASP computations, has dispersion been considered?

The correct abbreviation for Semicore Pseudopotential method is "SPP", not "DSPP"

Is "DS-PAW" is just the PAW method in VASP? When VASP was used, specify the plane-wave energy cutoff and pseudopotential choices. For example, we can say, "VASP calculations employed Semicore Pseudopotentials (SPP) and the Projector Augmented-Wave (PAW) method for core-electron interactions, and plane-wave energy cutoff was set to 450 eV" k-point sampling is needed for VASP

4. In Page 28, $E_{\text{bind}} = E_{\text{complex}} - (E_{\text{partA}} + E_{\text{partB}})$, while Figure 5 (a)-(c) used EF. Avoid using EF, in many cases people use it for Fermi energy.

5. It is recommended to explicitly compute the activation energies for nitrobenzene reduction, giving the transition states for $\text{NO}_2 \rightarrow \text{NHOH} \rightarrow \text{NH}_2$. Besides single-step pathways, also check if there is any possible multi-step reaction pathways.

Version 1:

Reviewer comments:

Reviewer #1

(Remarks to the Author)

The revised version can be accepted.

Reviewer #2

(Remarks to the Author)

After the revisions done by the authors, there are still issues in the computational study (see below) that prevent the publication of the manuscript as it is:

1) The authors did not address the comment on the model charge and on whether the ligands attached to Zn were deprotonated (O/NH) or protonated (OH/NH₂). The authors present experimental results supporting the idea that OH and NH₂ groups are protonated, likely because most of them are not bonded to Zn and therefore likely to be protonated. However, their computational model has one OH and two NH₂ groups deprotonated, and thus one O and one NH bonded to Zn (see xyz structure Cat-1b). If this is correct, the model is anionic, which could be possible under basic conditions. However, in that case, the support would deprotonate the alcohol, which has not been considered in the study.

2) By analysing the XYZ structures, I also observed that the hydride intermediate (Int-2b) has a formate group that detaches from Zn (Zn-O distance = ca 3 Å). However, there is no formate group in Int-1b, only CH₂CO₂ groups, indicating that one H atom appears to be missing. This means that all structures require a careful revision.

3) Concerning the MH vs MH₂ mechanisms (Figure R9), the MH₂ requires more detail. When an alcohol is dehydrogenated, it provides a H⁺ and H⁻, which could yield H₂. This can only form two metal hydrides after oxidative addition, which cannot take place in Zn²⁺. I also could not see the direct formation of MH₂ from an alcohol in the reference provided by the authors (J Am Chem Soc 2024, 146, 20518).

Reviewer #3

(Remarks to the Author)

Reviewer #4

(Remarks to the Author)

The authors significantly improved the manuscript, especially the clarity of the experimental methods used. Most of the concerns in the previous report have been addressed.

The following issues are to be considered before acceptance.

1. In the previous report, it stated "It is recommended to explicitly compute the activation energies for nitrobenzene reduction, giving the transition states for $\text{NO}_2 \rightarrow \text{NHOH} \rightarrow \text{NH}_2$. Besides single-step pathways, also check if there is any possible multi-step reaction pathways."

The authors performed detailed computations to address this issue, and concluded "Path 1 was thermodynamically more favorable and likely the dominant reaction pathway." Is there any experimental evidence to support this assumption? In the revised Supporting Information, it states, "Since we observed the presence of azobenzene (Ph-N=N-Ph) in the reaction mixture (Figure 4b in the manuscript) in the GC spectra, therefore, the proposed Path 2 was presented in below." It seems that the experimental finding supports Path 2.

2. More careful editing of the manuscript. Below are some examples.

(1) "As a control, in the reactions of benzyl alcohol with nitrobenzene and benzyl alcohol with aniline (steps 5-6), which all successfully yielded the target products with good yields after 21 h."

it is a fragment, and lacks a main clause (i.e., a complete sentence with a subject and verb that can stand alone). It may be corrected to "As a control, the reactions of benzyl alcohol with nitrobenzene and benzyl alcohol with aniline (steps 5-6)

were conducted, which both successfully yielded the target products with good yields after 21 h.”

(2) “2.4 Study on the DFT calculation” is wordy. “DFT calculations” is good enough.

(3) “As shown in Figures 5a-5c, density functional theory (DFT) was employed to optimize the structures and calculate the binding energies of ZnN₄-CS, ZnN₃O-CS, and ZnN₂O₂-CS”

The acronym, DFT, was defined before, thus there is no need to give “density functional theory (DFT)”

Version 2:

Reviewer comments:

Reviewer #4

(Remarks to the Author)

The authors well addressed my concerns in my previous report, publication is recommended.

Reviewer #5

(Remarks to the Author)

In my opinion, the author has made corresponding modifications based on the comments of Reviewer 2. I recommend this manuscript to be published in Nature Communications as it is.

Dear reviewers,

We thank reviewers for their positive comments and suggestions. The comments are highly valuable, and will largely improve our manuscript for meeting the high standard of *Nature Communications*. According to the comments, we have revised carefully our manuscript, and the revised parts were highlighted in blue. All the questions proposed by the reviewers have been answered point by point, and the comments have been explained sincerely as follows:

Response to the reviewers

To Reviewer 1:

This manuscript reported for the first time a novel bio-inspired Zn SACs with asymmetric Zn-N₂O₂ sites by utilizing biomass chitosan (Zn/CS), and achieved the first instance of heterogeneous Zn SACs in borrowing hydrogen reaction between nitroarenes and alcohols. Notably, the Zn/CS catalyst exhibited the highest TOF among various heterogeneous catalysts, as well as broad substrate scopes and excellent reusability, and the catalytic mechanism was revealed clearly. This work has significant original innovation, and is well written. It will guide researchers to find more interesting and efficient catalysts for the *N*-alkylation reaction. Therefore, I recommend this study for publication in *Nature Communications* with minor revision. To improve this work, I have the following comments and suggestions:

Response: Thanks for your support.

(1) Why was chitosan selected from various available biomasses for this study? The formation mechanism of chitosan microspheres should be elucidated.

Response: Thanks for your comment. We chose chitosan as our catalyst carrier for the following reasons: 1) Chitosan is a derivative of chitin, the world's second largest natural biomass resource; 2) Due to the rigidity of chitosan molecular chains, chitosan itself has a multi-level micro-nano porous structure, and its micro-nano structure can be further restructured through the sol-gel strategy by regulating hydrogen bonds. The multi-level micro-nano porous structure is conducive to the adhesion and dispersion of nano-metals and the diffusion of reaction substrates; 3) Its thermal/chemical

stability is good, which can resist alkali and various organic solvents; 4) The molecular chains of chitosan are rich in hydroxyl, amine and acetylamino groups, which can coordinate with various metal ions; especially the nitrogen atoms with high electronegativity, which contributes to the anchor of nano-metals and the electron transfer between metal-carrier interfaces, can significantly change the performance of the catalyst. Based on the above advantages of chitosan, we chose it as the support material for our catalyst.

The formation mechanism of chitosan microspheres was as follows: firstly, chitosan was completely dissolved in a LiOH/KOH/urea aqueous system under low temperature conditions to disrupt its original disordered micro-nanostructure. Then, guided by a surfactant, the sol-gel strategy was employed, and the "bottom-up" self-assembly process promoted the reorganization of chitosan molecules, ultimately leading to the formation of chitosan microspheres with a regular and uniform morphology.

(2) The Zn/CS catalyst was activated at 250 °C in an argon atmosphere from the Zn²⁺/CS precursor. Why was the Zn²⁺/CS precursor catalyst not used directly for catalysis?

Response: Thanks for your valuable comment. Based on your suggestion, we employed the Zn²⁺/CS precursor for the borrowing hydrogen reaction between nitrobenzene and benzyl alcohol. Under the same reaction conditions, the desired product yield was only 32% by using this precursor catalyst. Consequently, the catalyst was subjected to an activation treatment under an Ar atmosphere at 250 °C to modify the coordination environment and surface electronic structure around the Zn species, thereby enhancing its catalytic performance.

(3) In this study, petroleum ether was used as the reaction solvent. Had the authors tried using mixed solvents? A large number of literatures suggested that mixed solvents could have an unexpected promoting effect in organic reactions.

Response: We greatly appreciate your valuable suggestion. Based on your advice, we tried several typical mixed solvents in the model reaction of nitrobenzene and benzyl alcohol. As shown in Table R1, these mixed solvents included combinations of solvents with varying polarities, such as strong polarity, medium polarity and weak polarity. However, the experimental results indicated that none of these solvent combinations demonstrated superior catalytic performance compared to petroleum ether in this reaction. Based on this finding, we concluded that petroleum ether provided a better solvent environment in this reaction system, likely due to its solubility characteristics and solvation effects. Following your suggestion, we will focus on exploring the application of green solvents in future studies to develop more environmentally friendly and efficient solvent systems.

Table R1. The effects of mixed solvent on the borrowing hydrogen reaction of nitrobenzene with benzyl alcohol.

Entry	Catalyst	Solvent (v/v=1:1)	Yield ^{a,b} (%)
1	Zn/CS	Petroleum ether: Isooctane	25
2	Zn/CS	Petroleum ether: Toluene	NR
3	Zn/CS	Petroleum ether: Ethyl acetate	NR
4	Zn/CS	Petroleum ether: H ₂ O	NR
5	Zn/CS	Petroleum ether: Ethyl Alcohol	NR
6	Zn/CS	Petroleum ether: Dichloromethane	Trace
7	Zn/CS	Petroleum ether: N,N -dimethylformamide	NR

^a Reaction conditions: nitrobenzene (0.2 mmol), benzyl alcohol (0.6 mmol), KOH (0.54 mmol), solvent (4 mL), Zn/CS (0.09 mol% [Zn], Zn: PhNO₂), reacted at 120 °C for 21 h. ^b The yield of *N*-benzylaniline was determined by gas chromatography.

(4) Please provide data on nano-Zn/chitosan catalyst for the catalysis of benzyl alcohol and nitrobenzene, and compare it with the catalytic activity of the synthetic Zn SACs.

Response: We greatly appreciate your valuable advice. Based on your suggestion, we conducted a comparative study on the catalytic performance of nano-Zn/CS and single atom Zn/CS catalysts in the borrowing hydrogen reaction of nitrobenzene and benzyl alcohol. The experimental results indicated that the target yield of the nano-Zn/CS catalyst was only 46% in the same reaction condition, which was significantly lower than that of our Zn/CS catalyst. Additionally, we included the TEM image of the nano-Zn/CS catalyst, which exhibited nano structural features with an average Zn nanoparticle size of 2.36 nm. We attribute the lower catalytic activity of the nano-Zn/CS catalyst to the fact that, unlike single atom catalysts, it does not achieve full utilization of its catalytic Zn sites, leading to the reduced reaction efficiency. Based on your suggestion, we have added these data to the revised manuscript (see Figure S30).

Figure R1. TEM image and particle size distribution of nano-Zn/CS.

(5) The authors stated that the yield of N, N-dibenzylaniline decreased when increasing the catalyst amount. But there was no evidence, it should be provided.

Response: Thanks for your good advice. Based on the experimental results, when the catalyst dosage was 0.09 mol%, the target product yield reached a maximum of 98%. However, when the catalyst dosage was further increased to 0.12 mol%, the yield of

the target product significantly decreased. We speculated that the excessive amount of catalyst might have promoted side reactions, leading to the further conversion of some product into the side product of *N, N*-dibenzylaniline. To validate this hypothesis, we analyzed the reactants in the mixture with a higher catalyst amount, and successfully detected the side product of *N, N*-dibenzylaniline. Subsequently, we performed structural characterization of this product by using ^1H NMR and GC spectra. The results shown in Figures R2 and R3 clearly confirmed the structure of *N, N*-dibenzylaniline, further supporting our hypothesis.

Figure R2. ^1H NMR of *N, N*-dibenzylaniline.

Figure R3. GC diagram of *N, N*-dibenzylaniline.

(6) The substrates of aromatic alcohols were studied. What about the aliphatic alcohols with chain or ring structures in the substrate expansion? Such attempts should be included in the manuscript.

Response: We greatly appreciate your valuable advice. In fact, we had already tested various aromatic and heterocyclic nitro compounds or alcohol substrates, including electron-withdrawing and electron-donating groups, ortho/meta/para-substituents, halogenated compounds, and several important compounds with drug scaffolds in our preliminary work. Based on your suggestion, we further expanded the substrate scope and investigated several aliphatic alcohol substrates. As shown in Table R2, in the higher temperature of 140°C and larger catalyst dosage, the Zn/CS catalyst exhibited good catalytic performance in the *N*-alkylation reactions of some aliphatic alcohols, with relatively high yields of the target products. It clearly demonstrates that the Zn/CS catalyst is not only applicable to aromatic alcohol substrates but also possesses good catalytic activity in the *N*-alkylation reactions of aliphatic alcohols.

Table R2. *N*-alkylation of aliphatic alcohols catalyzed by Zn/CS.

$\text{R}_1\text{-NO}_2 + \text{R}_2\text{-CH}_2\text{OH} \xrightarrow{\text{Zn/CS}} \text{R}_2\text{-CH}_2\text{-NH-R}_1$		
Entry	Product	Yield (%)
1		85% ^a
2		84% ^a
3		82% ^a
4		86% ^b
5		96% ^c

^a Reaction conditions: nitro compounds (0.3 mmol), alcohol (0.9 mmol), KOH (1 mmol), petroleum ether (6 mL), Zn/CS (0.26 mol% [Zn], Zn: PhNO₂), reacted at 140 °C for 36 h. ^b toluene (4 mL), reacted at 140 °C for 21 h. ^c reacted at 140 °C for 21 h. Yields of the products were the separation yield, and the corresponding NMR spectra were displayed in supporting information.

To Reviewer 2:

In this work, a Zn single-atom catalyst based on chitosan support is used for the *N*-alkylation of nitrobenzenes with alcohols showing high activity and broad substrate scope under mild conditions, which is uncommon using non-noble metal SACs. The work also includes a mechanistic study that agrees with a hydrogen borrowing reaction (several steps have been isolated). DFT calculations are also included for one reaction step. While the experimental results are sound, the computational study and mechanistic proposal have several inconsistencies that require major revisions (see details). The computational Figure is also hard to read.

Response: Thanks for your support. We have carefully revised our manuscript according to your comments and suggestions, and also given cautious response to them.

(1) The initial models seem neutral, which could correspond to a Zn^{2+} neutralized by deprotonated NH_2 or OH groups after reaction with the base. However, this is not mentioned in the text, and it is very difficult to see in Figure 5 the presence of OH or O groups. In addition, different models can be constructed by changing the deprotonated groups. Has this been considered?

How does the selected model fit the experimental data? eg EXAFS results in Table S2 (e.g., Zn-N/O distance 1.99 Å) or the WT maximum value indicating the existence of dominant Zn-N bonds.

Response: Thanks for your valuable comments. To investigate whether the base (KOH) in the reaction mixture deprotonated the -OH/- NH_2 groups in chitosan support, FT-IR and C/H/N/O elemental analysis were supplemented for the Zn/CS catalyst before and after the reaction. In the FT-IR spectra (Figure R1), we specifically focused on the characteristic absorption peaks of -OH and - NH_2 . The results showed that the position and intensity of the -OH/- NH_2 peaks in Zn/CS almost unchanged before and after the reaction, indicating that the chitosan support did not undergo deprotonation during the reaction. Elemental analysis (Table R1) revealed that the C, H, O, N element contents showed no significant variation for the Zn/CS catalyst before and after the reaction, especially the changes in the content of H element. This

further supported the conclusion that the chitosan support was not deprotonated by the base. To further exclude the possibility of base-induced deprotonation of the chitosan support during the reaction, we also performed Zeta potential experiment on Zn/CS under identical base condition (0.54 mmol). As shown in below (Figure R2), the Zeta potential of the Zn/CS catalyst was close to zero, indicating that the Zn/CS catalyst exhibited electro-neutral characteristic under basic condition, suggesting that the -OH/-NH₂ groups in chitosan were not deprotonated in the presence of the base. Based on the above experimental results, we concluded that the -NH₂ and -OH functional groups in chitosan were not deprotonated by the base in the reaction mixture, but will anchor the Zn sites through coordination.

Figure R1. FT-IR spectra of the Zn/CS catalyst before and after the reaction.

Table R1. Elemental analysis of the Zn/CS catalyst before and after the reaction.

Sample	H %	N %	O %	C %
Zn/CS (Before the reaction)	4.23	8.41	45.25	42.11
Zn/CS (After the reaction)	4.29	8.50	45.04	42.17

Figure R2. The variation curve of Zeta potential of Zn/CS catalyst with time in the presence of KOH.

Regarding your comment about the difficulty in observing the -OH/NH₂ groups and the hard reading of computational Figure 5, we revised the computational Figure 5 and also optimized the figure by enhancing the visualization of the ligand atoms (especially the O/N atoms) to improve the clarity and readability. Given that the series of experiments mentioned above have ruled out the possibility of base-induced deprotonation of the chitosan support, we did not consider the deprotonation model of chitosan support in the theoretical calculations.

By integrating the experimental data and theoretical calculations, we conducted a detailed analysis of the Zn/CS catalyst structure to assess the compatibility between the selected model and experimental data. Firstly, the EXAFS data (Figure 2 and Table S2) clearly showed the presence of Zn–O/N coordination, with an average bond length of approximately 1.99 Å, which was consistent with typical Zn–N or Zn–O bond length. Wavelet transform (WT) analysis further confirmed the coexistence of Zn–N and Zn–O bonds in the catalyst structure, especially the dominant Zn–N bond. Meanwhile, XPS spectra of O1s and N1s for the pure chitosan and Zn/CS catalyst in Figures 1f-1g further confirmed the interaction between N/O elements and Zn atoms (i.e., the coordination of Zn–N and Zn–O).

These experimental results provided important guidance for our theoretical model.

Based on the coordination number of 4 for Zn–O/N and the Zn–N bond that cannot be ignored, we performed density functional theory (DFT) calculation and hypothesized three plausible coordination models: Zn–N₄, Zn–N₃O₁, and Zn–N₂O₂. The structural modeling results indicated that compared with other coordination models, especially the Zn–N₄ model, the Zn–N₂O₂ model had the smallest spatial constraints (Figures 5a–5c), which made it easier for the Zn active sites to fully contact the reaction substrates, facilitating the catalytic reaction. This was consistent with the excellent catalytic performance of the Zn/CS catalyst in our experimental results. Furthermore, as shown in Figures 5a–5c in the manuscript, the Zn–N₂O₂ model structure showed the highest binding energy (-6.19 eV), indicating the optimal thermodynamic stability. Therefore, Zn–N₂O₂ was chosen as the best coordination model. Overall, the selected Zn–N₂O₂ model fully combines experimental data and theoretical calculations, supporting its use as the representative structure of the catalyst. Based on your suggestion, we have revised some of the descriptions in our manuscript to make it clearer that the selected model fits the experimental data.

(2) If the support is deprotonated, the statement on page 15 mentioning “the NH₂ and OH groups from chitosan is the reason for the catalyst stability” should be corrected.

Response: Thanks for your suggestion. Based on the above experimental data, we demonstrated that the chitosan support was not be deprotonated. The -NH₂ and -OH functional groups in chitosan could stabilize the Zn active sites through coordination, effectively preventing the leaching and aggregation of Zn particles, thereby enhancing the stability, durability, and reusability of the catalyst.

(3) The same section mentions that the electron flow between chitosan and Zn atoms enhances the interaction between Zn and the support. This suggests that the support reduces the Zn atoms. Is there any support for this statement?

Response: Thanks for your valuable comment. As detailed in the manuscript, the chitosan support with abundant functional groups such as -NH₂ and -OH groups were capable of coordinating with Zn species to form Zn–N/O bonds, as confirmed by the

XPS and EXAFS spectra. To further prove the electron flow between chitosan and Zn atoms, N 1s and O 1s of XPS were conducted. As shown in Figures 1f-1g in the manuscript, after the loading of Zn atoms, some peaks attributed to N 1s or O 1s in Zn/CS shifted to lower binding energies compared to the pure calcined chitosan support, indicating that electrons flowed from Zn atoms to the chitosan support. We believe that the relatively higher electronegativity of N/O elements compared to Zn may be the reason that the electrons transfer from Zn to the N/O atoms in chitosan.

Not only that, we further supplemented the Zn 2p data of precursor Zn²⁺/CS catalyst. As shown in below, compared to precursor Zn²⁺/CS, the binding energies of Zn 2p for Zn/CS shifted to higher values, further proving that the electrons transferred from Zn to the N/O atoms in chitosan (Figure R3).

Figure R3. XPS spectra of Zn 2p for the Zn²⁺/CS and Zn/CS catalysts.

Moreover, the charge density difference analysis showed that the electron density of Zn atoms in Zn/CS significantly decreased, while a noticeable electron accumulation was observed around the chitosan support (Figure 5d). Bader charge analysis further visually presented the direction and quantity of the electron transfer between Zn atoms and chitosan support, and the chitosan support exhibited a tendency to gain electrons (Figure 5e). These results clearly demonstrated that the chitosan support coordinated with Zn through Zn-N/O interactions, with electrons flowing from the Zn atoms to the chitosan support. Compared to simple physical

adsorption, we believed that the coordination bonds (Zn-N/O) formed by the flow of electrons at the metal-support interface could significantly enhance the interaction force at the metal-support interface and improve the stability of the catalyst. As presented in the manuscript, the Zn/CS catalyst indeed exhibited excellent cyclic stability (Figures S13-S18). Based on your suggestion, we have added some of the above descriptions to the revised manuscript.

(4) For the Bader analysis and the computed energy pathway, it would be helpful to compare the results with a similar model, which is not reactive (e.g., Zn (NO₃)₂), instead of a Zn nanoparticle, which is more challenging to model. Is there any support for the size and shape of the nanoparticle used in the calculations?

Response: Thanks for your careful review. To highlight the advantages of Zn SACs (Zn single atoms supported on CS), we introduced the nano-Zn/CS catalyst (nano Zn supported on CS) as a comparison in the theoretical calculations. As shown in Figure R4a, the nano-Zn/CS catalyst demonstrated a significant activity advantage compared to other control catalysts in the *N*-alkylation of nitrobenzene and benzyl alcohol. Additionally, when compared to the Zn(NO₃)₂ catalyst, the nano-Zn/CS catalyst also exhibited higher target product yield within 24 h (Figure R4b).

We chose the nano-Zn/CS catalyst as the comparison for the theoretical calculations based on the following considerations: 1) In the comparative experiments with multiple catalysts, the catalytic activity of nano-Zn/CS was superior to that of other control catalysts; 2) The nano-Zn/CS model was more similar to our Zn/CS, as it had the same CS carrier, which could also highlight the advantages and necessity of synthesizing Zn single atoms.

Based on your suggestion, we supplemented the TEM image of the nano-Zn/CS catalyst. As shown in Figure R5, it exhibited nano structural features with an average Zn nanoparticle size of 2.36 nm. The TEM image of nano-Zn/CS have been added to the revised manuscript (Page 23 in the manuscript; Figure S30 in the supporting information).

Figure R4. Kinetic curves for various catalysts in the model reaction of nitrobenzene and benzyl alcohol.

Figure R5. TEM image and particle size distribution of the nano-Zn/CS catalyst.

(5) The energy profile assumes that the base reacts with the alcohol substrate instead of the support. Could the support deprotonate the alcohol instead of the base?

Response: Thanks for your valuable comment. To further investigate the crucial role of the base (KOH) in the reaction mixture and to verify whether the chitosan support could deprotonate the alcohol, we conducted control experiments without the addition of KOH in the presence of Zn/CS. As shown in Figure R6, under the condition without KOH, apart from the peaks of the reaction substrates benzyl alcohol and nitrobenzene, no new signal peaks of intermediates such as benzaldehyde, aniline, imine, *etc.* were observed in the GC spectra. Similarly, in the NMR spectra, only the

characteristic peak of substrate benzyl alcohol was observed. These data indicated that without the addition of KOH, the benzyl alcohol could not be deprotonated and further combined with the active Zn metal sites to undergo oxidation reaction to form the intermediate of benzaldehyde. As a comparison, in the experiments of our manuscript, after the addition of KOH, the benzyl alcohol was gradually consumed and oxidized to benzaldehyde, as observed in both the NMR and GC spectra (Figures 4a-4b in the manuscript).

Additionally, we systematically examined the effect of different amounts of KOH on the catalytic activity (Table S5 in the Supporting Information). The results indicated that as the amount of KOH increased, the yield of the target product significantly improved, indicating a positive correlation between base concentration and catalytic efficiency, further highlighting the indispensable role of the base in alcohol deprotonation.

Figure R6. GC (a) and ¹H-NMR (b) spectrum for the *N*-alkylation reaction of nitrobenzene and benzyl alcohol without the addition of base (KOH).

(6) Another way to test the model used for the reaction would be to compute the hydrogenation of imine and show that it has a lower energy barrier than the dehydrogenation of the alcohol, as suggested experimentally.

Response: Thanks for your good advice. Based on your suggestion, we further

investigated the changes in Gibbs free energy during the hydrogenation of imine. As shown in Figures R7-R8, the imine (E) first adsorbed onto the intermediate of Int-3b, forming the intermediate of Int-4b (step f). Subsequently, the Zn-H in Int-3b further combined with the C atom in C=N bond of imine to form the transition state of TS-2b, and the energy barrier that needed to be overcome was 0.22 eV (step g). Then, the TS-2b was hydrogenated to the intermediate of Int-5b. In subsequent steps, Int-5b continued to react with benzyl alcohol to form intermediates of Int-6b and Int-7b, ultimately producing the product of *N*-benzylaniline (F). It can be seen that during the hydrogenation of imine, the energy barrier that needed to be overcome for the formation of the transition state TS-2b was 0.22 eV, which was lower than the energy barrier of 0.25 eV that needed to be overcome for the oxidation of benzyl alcohol to benzaldehyde (in manuscript of Figure 5g).

Since we had proved that the oxidation of benzyl alcohol was the rate-determining step of the entire reaction, in the manuscript, we mainly presented the reaction pathway and corresponding Gibbs free energy profiles for benzyl alcohol oxidation by using the Zn/CS and nano-Zn/CS catalysts. Based on your suggestion, we have added the reaction pathway and corresponding Gibbs free energy profiles for hydrogenation of imine in the revised manuscript (Page 25) and supporting information (Figure S25-S26).

Figure R7. The proposed mechanism for the borrowing hydrogen reaction of nitroarenes with alcohols catalyzed by Zn/CS.

Figure R8. The reaction pathway and corresponding Gibbs free energy profiles for hydrogenation of imine by using the Zn/CS catalyst.

(7) The discussion on deuterium labeling on page 19 is unclear. The same is true for the equations in Figure 4 d1 and d2. This should be clarified in the text and with some more explicit schemes, either in the manuscript or in the supporting information.

Response: Thanks for your valuable comment. In the *N*-alkylation reaction of nitroarenes with alcohols via borrowing hydrogen strategy, two typical reaction pathways may be involved: the metal-hydride pathway (M-H) and the direct hydrogen

transfer pathway. In heterogeneous catalytic systems catalyzed by transition metals, the metal hydride pathway was typically considered as the main reaction mechanism [Coord. Chem. Rev. **2024**, 541, 216750; Chem. Rev. **2018**, 118, 1410]. Therefore, as shown in below, we proposed two different metal hydride reaction pathways: the monohydride pathway (M-H, path A) and the dihydride pathway (M-H₂, path B). In path A, the benzyl alcohol underwent deprotonation in the presence of base (KOH) and then connected with the Zn particle. The Zn particle was further connected with the α -C-H in Ar-CH₂-O⁻ to form Zn-H (i.e., M-H). In path A, the target product tended to be Ar-NH-CD/D-Ar. In contrast, Path B involved the O-H and α -C-H of the alcohol interacting with the Zn species to generate the Zn-H₂ species (i.e., M-H₂), which was more likely to produce the target product of Ar-NH/D-CH/D-Ar [J. Am. Chem. Soc. **2024**, 146, 20518; Angew. Chem. Int. Ed. **2019**, 58, 5417; Catal. Sci. Technol. **2020**, 10, 3458].

Figure R9. Two different pathways proposed for the metal hydride pathway.

To distinguish which reaction pathway was more likely in our catalytic mixture for the *N*-alkylation of nitrobenzene and benzyl alcohol, we conducted an isotope labeling experiment by using deuterated benzyl alcohol (α -C-H position) as the hydrogen source. In the ^1H NMR spectra (Figures S21-S22 and S24 in the supporting information), it detected the target product *N*-benzylaniline with types of C-H/H (1%), C-H/D (7%) and C-D/D (92%), and the N-H bond in *N*-benzylaniline showed no deuterated H atoms. High resolution mass spectrometry (HR-MS) analysis further confirmed the presence of *N*-benzylaniline with types of C-H/H, C-H/D and C-D/D, particularly the predominant C-D/D type (Figure S23 in the supporting information). These results indicated that our reaction mixture predominantly proceeded through the Zn-H pathway (Path A). Based on your suggestion, to make the deuterium labeling and the equations in Figures 4d₁ and 4d₂ easier for readers to understand, we have provided further clarification in the revised manuscript, as shown in Page 19 in the manuscript and Figures S20-S24 in the supporting information.

(8) On page 23, it is unclear what type of kinetic study has been done to conserve the computational resources. Figure S26 is not given in the supporting information.

Response: Thanks for your careful review. We apologize for our previous unclear expression. In the model reaction for *N*-alkylation of nitrobenzene and benzyl alcohol. As stated in the manuscript, the reaction proceeded through the following pathways: 1) benzyl alcohol was oxidized to benzaldehyde by the catalyst, generating the metal-H species; 2) the formed metal-H reduced the nitrobenzene to aniline; 3) the benzaldehyde coupled with aniline to form an imine intermediate; 4) the imine intermediate was further reduced by the metal-H to yield *N*-benzylaniline. In the initial version of the manuscript, we aimed to determine which pathway mentioned above was the rate-determining step (RDS) of the entire reaction, and focused only on studying the Gibbs free energy changes in the RDS reaction process to conserve computational resources.

Figure R10. The kinetic plots for nitrobenzene, benzyl alcohol, and *N*-benzylimine.

As shown in below, in the reactions involving nitrobenzene with benzyl alcohol, as well as *N*-benzylideneaniline with benzyl alcohol, both the reactions exhibited a positive association, indicating the first-order kinetics. Notably, compared to the conversion of nitrobenzene (represented the reaction of nitrobenzene to aniline) or the conversion of *N*-benzylideneaniline (represented the reaction of the imine intermediate to *N*-benzylaniline), the rate constant for the conversion of benzyl alcohol (represented the reaction of benzyl alcohol to benzaldehyde) was the smallest, suggesting that the dehydrogenation of benzyl alcohol constituted the RDS in the borrowing hydrogen reaction between nitrobenzene and benzyl alcohol. Based on your suggestion, we have added some of the above description to the revised manuscripts (see Page 23) and corrected Figure S31 in the revised supporting information. Herein, based on your advice, we also computed the key process of hydrogenation of imine (in question 6) and showed that it indeed had a lower energy

barrier than the dehydrogenation of the alcohol.

(9) On the same page, the sentence “The Gibbs free energy changes involved in this borrowing hydrogen reaction were investigated in combination with DFT simulations” is unclear.

Response: Thanks for your careful review. Sorry for our inappropriate expression. Based on your suggestion, we revised it as “The variation of Gibbs free energy in this borrowing hydrogen reaction was studied by using DFT simulation”.

(10) It is recommended that all computed structures in the manuscript be included in the supporting information as an xyz file to facilitate their visualization using different software.

Response: Thanks for your valuable advice. Following your suggestion, apart from the model information of Zn/CS and nano-Zn/CS presented in the supporting information, we have added all computed structures (xyz file) involved in the manuscript to a compressed file.

To Reviewer 3:

Response: Thank you for your valuable comments and support.

To Reviewer 4:

In this manuscript, Huang et al. reported the application of bio-inspired Zn single-atom catalysts (SACs) with asymmetric Zn-N₂O₂ sites for the borrowing hydrogen reaction between nitroarenes and alcohols. They found that the chitosan-derived support enhances Zn atom dispersion and electronic modulation, leading to exceptional catalytic activity with the highest turnover frequency among reported heterogeneous catalysts. Their DFT computations helped gain deeper insights.

In general, this work presents a new strategy for biomass-derived non-noble metal SACs, offering a highly efficient, green, and recyclable catalyst for N-alkylation reactions under mild conditions. I am counting on the experimental peers to evaluate the experimental sections. Below I am only commenting on the theoretical part.

The authors performed DFT computations to help reveal the underlying mechanism, specifically, they revealed that the electron-deficient Zn-N₂O₂ sites can facilitate the formation of key Zn-H and Zn-O intermediates, thus accelerating alcohol dehydrogenation, and Zn atoms serve as active centers for charge redistribution, enhancing catalytic efficiency. Generally, the computations helped understand/validate the experimental observations, and provide a predictive framework for designing more efficient single-atom catalysts for borrowing hydrogen reactions. However, more careful work should be done about the presentation and description of the computational methods and results.

Response: Thanks for your support.

(1) In the main text, it states “All density functional theory (DFT) calculations were performed using the DMol³ package, Vienna Ab-initio Simulation Package (VASP),

and Projector augmented-wave method (DS-PAW)” The authors should specify which software was used for which calculations, Or, if all calculations were done in one package, state explicitly which one.

Response: Thanks for your valuable comment. We sincerely apologize for the lack of clarity in the original description, which may have caused confusion. In response to your suggestion, we have thoroughly revised the computational details in both the main text and the Supporting Information to clearly specify the software used for each part of the calculation. In the revised manuscript: Geometric optimization, total energy calculation, and reaction pathway searches for the composite were carried out using the DMol³ software package. Bader charge analysis, density of states (DOS) and charge density difference calculations were performed using the Vienna Ab initio Simulation Package (VASP). The computational details are revised in the supporting information (see Page 31, part of Computational Details in the supporting information). We have copied the revised version about the detailed computational methods below in response 2.

(2) Since the computational details are given in the Supporting Information, for the section, S2. Computational Details, the description is also not clear. It seems that the complex was simulated by a molecular cluster model, most computations were done with DMol³ code, including geometry optimization, energetic evaluations and reaction pathway search. VASP code was only used to get Bader charge, density of states, and charge density difference for the complex, and a large supercell, (30x30x30) angstrom, was used to simulate cluster models. It seems that for Figure 5, the important figure about theoretical studies, d-f are by VASP, while others are by DMol³. If yes, clearly specify them in the figure caption and also in the main text. In Figure 5f, both total density of states and partial density of states are presented, but the figure is too small. A larger figure can be given in the Supporting Information, and some more discussions are helpful.

Response: Thanks for your careful review and helpful suggestions. As mentioned above, the geometric optimizations, energy calculations, and reaction pathway

searches were performed using the DMol³ software package. The Bader charge analysis, density of states and charge density difference were calculated using the VASP code.

To avoid artificial interactions due to periodic boundary conditions in the cluster model, a large cubic supercell of (30 × 30 × 30) Å³ was employed in the VASP calculations, including Bader charge analysis, DOS, and charge density difference. For these calculations, a 1 × 1 × 1 *k*-point grid was used, and Brillouin-zone sampling was performed using a Γ -centered Monkhorst–Pack scheme. In our revised version, we have revised the S2. Computational Details section in supporting information as follows:

S2. Computational Details

Geometry optimizations, total energy calculations, and reaction pathway searches of the composite were performed using the DMol³ software package¹. The generalized gradient approximation (GGA) with the Perdew-Burke-Ernzerhof (PBE) functional², along with the semi-core pseudopotential (SPP) method and double numerical basis sets plus polarization (DNP)³ were employed. The self-consistent field (SCF) convergence criterion for electronic energy was set to 1.0×10^{-5} Ha. Geometry optimization convergence thresholds were set to 1.0×10^{-5} Ha for energy, 0.004 Ha Å⁻¹ for force, and 0.01 Å for displacement. The binding energy was calculated using the formula: $E_{\text{bind}} = E_{\text{complex}} - (E_{\text{partA}} + E_{\text{partB}})$. The transition states were located using the linear synchronous transit (LST) and quadratic synchronous transit (QST) methods⁴, combined with conjugated gradient (CG) refinement. The Gibbs free energy (*G*) was obtained using the equation: $G = E_{\text{total}} + E_{\text{ZPE}} - TS$, where E_{total} was the ground-state electronic energy, E_{ZPE} was the zero-point energy, *T* was the temperature, and *S* was the entropy derived from vibrational frequency analysis. Finally, the Gibbs free energy change (ΔG) for each elementary reaction was defined as: $\Delta G = G_{\text{P}} - G_{\text{R}}$, where G_{P} was the total energy of all products and G_{R} was the total energy of all reactants.

The Bader charge analysis, density of states (DOS), and charge density difference calculations were carried out using the Vienna Ab initio Simulation Package (VASP)^{5,6}. The exchange-correlation functional was described by the GGA-PBE approach, and core-valence interactions were treated using the projector augmented-wave (PAW) method^{7,8}. A plane-wave energy cutoff of 450 eV was used. To model the isolated molecular clusters, all VASP calculations were performed in a $(30 \times 30 \times 30)$ Å³ cubic simulation cell to avoid spurious interactions between periodic images. Brillouin-zone sampling was carried out using a Γ -centered Monkhorst-Pack scheme with a $1 \times 1 \times 1$ k -point mesh. Dispersion interactions were included using the DFT-D3 correction method proposed by Grimme et al.⁹, ensuring an accurate description of long-range van der Waals interactions.

In response to your suggestion, we have clearly indicated the software used for each part of the study in both the main text and the caption of Figure 5 (see Page 26). Specifically, Figures 5d-5f were obtained using VASP, while the other panels were generated using DMol³. Additionally, we acknowledged that the total and partial DOS plots in Figure 5f were too small. Based on your advice, we have provided an enlarged version of the figure in the revised supporting information (see Figure S29), and expanded the corresponding discussion in the main text to provide more insight, as follows: “As shown in Figure 5f and S29, the incorporation of Zn atoms introduced new electronic states near the Fermi level, indicating a significant modification of the local electronic structure of the chitosan support. The partial DOS revealed that these new states involved Zn 3d and the neighboring N/O atoms' orbitals, pointing to strong orbital hybridization between Zn and the N/O atoms in chitosan, confirming the successful coordination of Zn. Consequently, the electronic conductivity of the catalyst could be enhanced. The narrowed bandgap and increased DOS near the Fermi level indicated the improved charge-carrier mobility. The modification of the electronic structure was expected to facilitate the electron transfer during catalytic reactions, thereby enhancing the catalyst's activity” (see Pages 22-23 in the revised manuscript). We sincerely appreciate your suggestions, which have helped us improve

both the clarity and scientific depth of our theoretical analysis.

(3) For VASP computations, has dispersion been considered? The correct abbreviation for Semicore Pseudopotential method is "SPP", not "DSPP". Is "DS-PAW" is just the PAW method in VASP? When VASP was used, specify the plane-wave energy cutoff and pseudopotential choices. For example, we can say, "VASP calculations employed Semicore Pseudopotentials (SPP) and the Projector Augmented-Wave (PAW) method for core-electron interactions, and plane-wave energy cutoff was set to 450 eV" k-point sampling is needed for VASP

Response: Thanks for your valuable and detailed suggestions regarding the computational details, especially for the VASP calculations. We have revised the manuscript accordingly to address all of your concerns:

1. Dispersion interactions:

Yes, dispersion effects were considered in the VASP calculations. Specifically, we employed the DFT-D3 correction method proposed by Grimme et al. to accurately account for long-range van der Waals interactions. This has now been clearly stated in the revised computational details.

2. Pseudopotential abbreviations:

Thanks for pointing out the incorrect abbreviation. We have corrected the description of the semi-core pseudopotential method from "DSPP" to the proper "SPP" in the DMol³ -related part.

3. Clarification of PAW method:

The term "DS-PAW" previously mentioned was a misnomer. It has been corrected to the standard "PAW" (Projector Augmented-Wave) method, as implemented in the VASP code. This correction has been applied consistently throughout the main text and the Supporting Information.

4. VASP calculation settings:

In the revised S2. Computational Details section, we have explicitly provided the key settings used in the VASP calculations, including:

- ✓ Use of the PAW method for core-valence interactions
- ✓ Exchange-correlation functional described by GGA-PBE

- ✓ Plane-wave energy cutoff set to 450 eV
- ✓ All calculations performed in a $(30 \times 30 \times 30) \text{ \AA}^3$ cubic supercell to eliminate spurious periodic interactions
- ✓ Brillouin-zone sampling using a Γ -centered Monkhorst-Pack scheme with a $1 \times 1 \times 1$ k -point mesh.

These modifications are now fully reflected in the revised S2. Computational Details section in the Supporting Information, and the corresponding descriptions in the main text have also been updated for consistency. We believe these clarifications comprehensively address all points raised and enhance the transparency and reproducibility of our computational study.

(4) In Page 28, $E_{\text{bind}} = E_{\text{complex}} - (E_{\text{partA}} + E_{\text{partB}})$, while Figure 5 (a)-(c) used E_{F} . Avoid using E_{F} , in many cases people use it for Fermi energy.

Response: Thanks for your careful review. We noted that the binding energy was denoted as E_{bind} in the computational formulas, but E_{F} was incorrectly used for labeling in Figures 5(a)-(c). To avoid confusion, based on your suggestion, we revised all relevant labels in the revised manuscript to E_{bind} .

(5) It is recommended to explicitly compute the activation energies for nitrobenzene reduction, giving the transition states for $\text{NO}_2 \rightarrow \text{NHOH} \rightarrow \text{NH}_2$. Besides single-step pathways, also check if there is any possible multi-step reaction pathways.

Response: Thanks for your valuable comment. Based on your suggestion, we computed the activation energies for the reduction of nitrobenzene, considering the transition states for $\text{NO}_2 \rightarrow \text{NHOH} \rightarrow \text{NH}_2$ process. As shown in below (Path 1), the H^* represented the Zn/CS-H (i.e., the Int-3b in our manuscript). Firstly, the $-\text{NO}_2$ in nitrobenzene connected with the Zn-H in Int-3b to form intermediates Int-1c and Int-2c. Then, the H atom in Zn-H of Int-3b was added to nitrobenzene to form the transition state of TS-1c, which required overcoming an energy barrier of 0.15 eV. Subsequently, after the dehydration reaction from TS-1c, intermediate of Ph- NO^* (Int-3c) was formed, which then connected with Int-3b to form intermediate of Int-4c.

The intermediate of Int-4c underwent hydrogenation with the H atom in Zn-H of Int-3b to form intermediate of Int-5c, then leading to the formation of intermediate Ph-NHOH* (Int-6c). The intermediate of Int-6c was then adsorbed onto Zn-H of Int-3b to form the transition state of TS-2c (with overcoming an energy barrier of 0.11 eV), which was further underwent dehydration to form the intermediate of Int-7c. Finally, the Int-7c connected with Zn-H in Int-3b to form intermediate of Int-8c, which was further hydrogenated to form intermediate of Int-9c, ultimately yielding the product aniline.

Since we observed the presence of azobenzene (Ph-N=N-Ph) in the reaction mixture (Figure 4b in the manuscript) in the GC spectra, therefore, the proposed Path 2 was presented in below. The intermediates of nitrosobenzene* (Ph-NO*) and N-phenylhydroxylamine* (Ph-NHOH*) could directly react to form intermediate of Int-1d, with the removal of H₂O. The Int-1d could bound with Zn-H in Int-3b to form intermediate of Int-2d and Int-3d. The Int-3d further underwent hydrogenation and dehydration to form intermediate of Int-4d. The intermediate of Int-4d was progressively hydrogenated to form intermediates of Int-5d and Int-6d, respectively, and the Int-6d was further transformed into intermediate of Int-7d. Finally, the Int-7d was hydrogenated to Int-8d, and ultimately yielding the product aniline.

It could be seen that although the overall energy fluctuation of Path 2 was relatively small, the energy of the final product (-7.16 eV) was higher than that of Path 1. Therefore, although both paths can produce the target product, Path 1 was thermodynamically more favorable and likely the dominant reaction pathway. Based on your suggestion, we have added these data in the revised manuscript (Pages 25) and the supporting information (Figure S32).

Figure R1. The proposed mechanism for the nitrobenzene reduction to aniline catalyzed by Zn/CS, and the corresponding changes of Gibbs free energy.

References

- [1] Delley, B. From molecules to solids with the DMol³ approach. *J. Chem. Phys.* **113**, 7756-7764 (2000).
- [2] Perdew, J. P., Burke, K. & Ernzerhof, M. Generalized Gradient Approximation Made Simple. *Phys. Rev. Lett.* **77**, 3865-3868 (1996).
- [3] Delley, B. Hardness conserving semilocal pseudopotentials. *Phys. Rev. B.* **66**, 155125 (2002).
- [4] Govind, N., Petersen, M., Fitzgerald, G., King-Smith, D. & Andzelm, J. A generalized synchronous transit method for transition state location. *Comput. Mater. Sci.* **28**, 250-258 (2003).
- [5] Kresse, G. & Hafner, J. Ab initio molecular dynamics for liquid metals. *Phys. Rev. B.* **47**, 558-561 (1993).
- [6] Kresse, G. & Furthmüller, J. Efficiency of ab-initio total energy calculations for metals and semiconductors using a plane-wave basis set. *Comput. Mater. Sci.* **6**, 15-50 (1996).
- [7] Blochl, P., Blöchl, E. & Blöchl, P. E. Projected augmented-wave method. *Phys. Rev. B Condens. Matter.* **50**, 17953-17979 (1994).
- [8] Kresse, G. & Joubert, D. From ultrasoft pseudopotentials to the projector augmented-wave method. *Physical Review B.* **59**, 1758-1775 (1999).
- [9] Grimme, S., Antony, J., Ehrlich, S. & Krieg, H. A consistent and accurate ab initio parametrization of density functional dispersion correction (DFT-D) for the 94 elements H-Pu. *J. Chem. Phys.* **132**, 154104 (2010).

Dear reviewers,

We thank reviewers for their positive comments and suggestions. The comments are highly valuable, and will largely improve our manuscript for meeting the high standard of *Nature Communications*. According to the comments, we have revised carefully our manuscript, and the revised parts were highlighted in blue. All the questions proposed by the reviewers have been answered point by point, and the comments have been explained sincerely as follows:

Response to the reviewers

To Reviewer 1:

The revised version can be accepted

Response: Thanks for your support.

To Reviewer 2:

After the revisions done by the authors, there are still issues in the computational study (see below) that prevent the publication of the manuscript as it is:

Response: Thanks for your support. We have carefully revised our manuscript according to your comments and suggestions, and also given cautious response to them.

(1) The authors did not address the comment on the model charge and on whether the ligands attached to Zn were deprotonated (O/NH) or protonated (OH/NH₂). The authors present experimental results supporting the idea that OH and NH₂ groups are protonated, likely because most of them are not bonded to Zn and therefore likely to be protonated. However, their computational model has one OH and two NH₂ groups deprotonated, and thus one O and one NH bonded to Zn (see xyz structure Cat-1b). If this is correct, the model is anionic, which could be possible under basic conditions. However, in that case, the support would deprotonate the alcohol, which has not been considered in the study.

Response: Thanks for your valuable suggestion. Initially, our computational model assumed the deprotonation of one OH group and two NH₂ groups, based on the most stable coordination configuration identified during the optimization of the chitosan-Zn

interaction. However, the results of our current experimental investigations suggest that the OH and NH₂ groups are protonated, likely because most of them are not coordinated to Zn and therefore likely to be protonated (the Zn loading is only 1.11 wt%). Under basic conditions, those small amounts of deprotonated OH/NH₂ that coordinate with Zn atoms may indeed cause the model to be anionic. In this case, we conduct a series of additional experiments to verify whether the support would deprotonate the alcohol.

Time-resolved diffuse reflectance infrared Fourier transform spectroscopy (DRIFTS) was first performed. As shown in Figure R1a, in the absence of an external base, the C-O stretching vibration peak of benzyl alcohol (1073 cm⁻¹) adsorbed on the Zn/CS catalyst showed negligible change over the course of 1 h. In contrast, the peak shifted significantly in the presence of KOH (Figure R1b), which was consistent with the formation of alkoxide intermediates (*J. Catal.* **2024**, *434*, 115537; *Appl. Catal. B Environ.* **2022**, *319*, 121904). This shift directly supports the hypothesis that the benzyl alcohol undergoes deprotonation under basic conditions. The comparison clearly demonstrates that the activation of benzyl alcohol is strongly dependent on the presence of an external base.

To further substantiate these findings, we conducted gas chromatography (GC) (Figure R2a), ¹H NMR (Figure R2b), and GC-MS (Figure R3) analyses. After 3 h of reaction, only the characteristic peaks of the starting substrates (benzyl alcohol and nitrobenzene) were detected, with no intermediate or oxidation product signals observed in the absence of KOH. These data confirm that the Zn/CS catalyst does not achieve the deprotonation and the subsequent conversion of benzyl alcohol in the absence of base. We attribute this to the weak basicity of the support, which is insufficient to induce alcohol deprotonation and the subsequent oxidation reaction. In contrast, the addition of KOH in the reaction mixture led to gradual conversion of both benzyl alcohol and nitrobenzene (Figures 4 in the manuscript). Therefore, the support cannot deprotonate the alcohol in the absence of KOH.

We sincerely appreciate your thorough review, which has significantly enhanced the scientific rigor of our work. In response, we have made the appropriate revisions and additions to the manuscript (Page 11 in the manuscript; Pages 10-11 in the supporting

information).

Figure R1. Time-resolved DRIFTS spectra of the benzyl alcohol oxidation process with the addition of Zn/CS catalyst without KOH (a) and Zn/CS catalyst with KOH (b).

Figure R2. GC (a) and ¹H NMR (b) spectra of the borrowing hydrogen reaction between nitrobenzene and benzyl alcohol in the presence of Zn/CS catalyst without KOH.

Figure R3. GC-MS spectrum of the borrowing hydrogen reaction between nitrobenzene and benzyl alcohol in the presence of Zn/CS catalyst without KOH.

(2) By analysing the XYZ structures, I also observed that the hydride intermediate (Int-2b) has a formate group that detaches from Zn (Zn-O distance = ca 3Å). However, there is no formate group in Int-1b, only CH₂CO₂ groups, indicating that one H atom appears to be missing. This means that all structures require a careful revision.

Response: We sincerely appreciated your careful review. We regret that, due to an oversight, some intermediates in our model were missing one hydrogen atom. To address this, we carefully re-examined all relevant computational structures and made the necessary corrections and updates. Additionally, we updated the corresponding XYZ structure files to accurately reflect these changes. Once again, we are grateful for your meticulous review, and we have made the required revisions, including updates to the corresponding models and structural descriptions.

(3) Concerning the MH vs MH₂ mechanisms (Figure R9), the MH₂ requires more detail. When an alcohol is dehydrogenated, it provides a H⁺ and H⁻, which could yield H₂. This can only form two metal hydrides after oxidative addition, which cannot take place in Zn²⁺. I also could not see the direct formation of MH₂ from an alcohol in the reference provided by the authors (J Am Chem Soc. 2024, 146, 20518).

Response: We sincerely appreciated your valuable feedback. Indeed, traditional MH₂ species were typically formed through oxidative addition at the metal center, a process that was thermodynamically and kinetically challenging for Zn²⁺. We fully agreed with this point and appreciated your prompt to reconsider and clarify the description of this key mechanism. The "MH₂ pathway" mentioned in the manuscript and supplementary materials followed the mechanistic classification commonly used in borrowing hydrogen or hydrogen transfer reactions [*Top. Curr. Chem.* **2016**, 374, 27]. In this reference, the distinction between "MH" and "MH₂" pathways was not strictly based on the oxidative addition capability of specific metals, but rather as a general, experimentally distinguishable classification:

(1) **In MH path**, the benzyl alcohol underwent deprotonation in the presence of base, followed by coordination to the metal (M) center. The metal particle was further connected with the α-C-H in Ar-CH₂-O⁻ to form M-H. In MH path, the target product

tended to be Ar-NH-CD/D-Ar.

(2) **In MH₂ path**, it involved the O-H and α -C-H of the alcohol interacting with the metal to generate the M-H₂ species, which was more likely to produce the target product of Ar-NH/D-CH/D-Ar.

Our deuterated experiments using PhCD₂OH as the substrate revealed that the distribution of D/H in the products followed the characteristic pattern of the MH pathway. The "MH₂ pathway" described in the literature [*J. Am. Chem. Soc.* **2024**, *146*, 20518] referred to this general mechanistic classification and was not intended to imply the actual formation of MH₂ species in our Zn SACs system. To avoid misleading the readers, we revised the corresponding descriptions based on your suggestion.

To Reviewer 3:

Response: Thanks for your valuable comments and support.

To Reviewer 4:

The authors significantly improved the manuscript, especially the clarity of the experimental methods used. Most of the concerns in the previous report have been addressed. The following issues are to be considered before acceptance.

Response: Thanks for your support.

(1) In the previous report, it stated “It is recommended to explicitly compute the activation energies for nitrobenzene reduction, giving the transition states for $\text{NO}_2 \rightarrow \text{NHOH} \rightarrow \text{NH}_2$. Besides single-step pathways, also check if there is any possible multi-step reaction pathways.” The authors performed detailed computations to address this issue, and concluded “Path 1 was thermodynamically more favorable and likely the dominant reaction pathway.” Is there any experimental evidence to support this assumption? In the revised Supporting Information, it states, “Since we observed the presence of azobenzene (Ph-N=N-Ph) in the reaction mixture (Figure 4b in the manuscript) in the GC spectra, therefore, the proposed Path 2 was presented in below.” It seems that the experimental finding supports Path 2.

Response: Thanks for your thorough review and constructive feedback. To further substantiate the dominant reaction pathway, we revisited our experimental data. The time-resolved ^1H NMR spectra (Figure 4a in the manuscript) indicated the presence of trace phenylhydroxylamine (Ph-NHOH) intermediate. However, due to its low concentration, this intermediate was not detected in the GC analysis (Figure 4b in the manuscript). As you noted, the GC kinetic spectra demonstrated the formation of substantial amounts of azobenzene (Ph-N=N-Ph) as the reaction progressed. Crucially, no characteristic peak for the Ph-NHOH intermediate was observed, suggesting that the

azobenzene formation might indeed be the primary pathway. To gain more accurate insight into the conversion of reactants, we further conducted GC-MS analysis. As shown in Figure R1, after 3 h of reaction, it detected a large amount of Ph-N=N-Ph, but without the Ph-NHOH intermediate. Therefore, the Path 2 was indeed the dominant reaction pathway. Moreover, the ΔG (Gibbs free energy change) was more advantageous for Path 2 (Page 24 in the supporting information), further proving that Path 2 was the main reaction pathway. Based on your suggestion, we have revised the corresponding description in the revised manuscript (Page 17-18 in the manuscript; Page 24-25 in the supporting information). We once again thank you for your meticulous review, which prompted us to refine our experimental analysis and avoid potential errors, ultimately enhancing the quality and reliability of our work.

Figure R1. GC-MS spectrum of the borrowing hydrogen reaction between nitrobenzene and benzyl alcohol at 3 h.

2 More careful editing of the manuscript. Below are some examples.

(1) “As a control, in the reactions of benzyl alcohol with nitrobenzene and benzyl alcohol with aniline (steps 5-6), which all successfully yielded the target products with good yields after 21 h.” it is a figment, and lacks a main clause (i.e., a complete sentence with a subject and verb that can stand alone). It may be corrected to “As a control, the reactions of benzyl alcohol with nitrobenzene and benzyl alcohol with aniline (steps 5–6) were conducted, which both successfully yielded the target products with good yields after 21 h.”

Response: Thanks for your good suggestion. Sorry for our poor description. Based on your suggestion, we have revised it as “As a control, the reactions of benzyl alcohol

with nitrobenzene and benzyl alcohol with aniline (steps 5–6) were conducted, which both successfully yielded the target products with good yields after 21 h.” in the revised manuscript.

(2) “2.4 Study on the DFT calculation” is wordy. “DFT calculations” is good enough.

Response: Thanks for your good advice. Based on your suggestion, we have revised it as “2.4 DFT calculations” in the revised manuscript.

(3) “As shown in Figures 5a-5c, density functional theory (DFT) was employed to optimize the structures and calculate the binding energies of ZnN₄-CS, ZnN₃O-CS, and ZnN₂O₂-CS” The acronym, DFT, was defined before, thus there is no need to give “density functional theory (DFT)”.

Response: We sincerely appreciated your meticulous review of the manuscript. Based on your suggestion, we have revised it “As shown in Figures 5a-5c, DFT was employed to optimize the structures and calculate the binding energies of ZnN₄-CS, ZnN₃O-CS, and ZnN₂O₂-CS” in the revised manuscript.